# Does a sparse ReLU network training problem always admit an optimum?

Quoc-Tung Le          Elisa Riccietti          Rémi Gribonval

Univ. Lyon, Inria, CNRS, ENS de Lyon, UCB Lyon 1, LIP UMR 5668, F-69342 Lyon, France

## Abstract

Given a training set, a loss function, and a neural network architecture, it is often taken for granted that optimal network parameters exist, and a common practice is to apply available optimization algorithms to search for them. In this work, we show that the existence of an optimal solution is not always guaranteed, especially in the context of *sparse* ReLU neural networks. In particular, we first show that optimization problems involving deep networks with certain sparsity patterns do not always have optimal parameters, and that optimization algorithms may then diverge. Via a new topological relation between sparse ReLU neural networks and their linear counterparts, we derive –using existing tools from real algebraic geometry– an algorithm to verify that a given sparsity pattern suffers from this issue. Then, the existence of a global optimum is proved for every concrete optimization problem involving a one-hidden-layer sparse ReLU neural network of output dimension one. Overall, the analysis is based on the investigation of two topological properties of the space of functions implementable as sparse ReLU neural networks: a best approximation property, and a closedness property, both in the uniform norm. This is studied both for (finite) domains corresponding to practical training on finite training sets, and for more general domains such as the unit cube. This allows us to provide conditions for the guaranteed existence of an optimum given a sparsity pattern. The results apply not only to several sparsity patterns proposed in recent works on network pruning/sparsification, but also to classical dense neural networks, including architectures not covered by existing results.

## 1   Introduction

The optimization phase in deep learning consists in minimizing an objective function w.r.t. the set of parameters $\theta$ of a neural network (NN). While it is arguably sufficient for optimization algorithms to find local minima in practice, training is also expected to achieve the infimum in many situations (for example, in overparameterized regimes networks are trained to zero learning error).

In this work, we take a step back and study a rather fundamental question: *Given a deep learning architecture possibly with sparsity constraints, does its corresponding optimization problem actually admit an optimal $\theta^*$?* The question is important for at least two reasons:

1. Practical viewpoint: If the problem does not admit an optimal solution, optimized parameters necessarily diverge to infinity to approximate the infimum (which always exists). This phenomenon has been studied thoroughly in previous works in other contexts such as tensor decomposition [7], robust principal component analysis [30], sparse matrix factorization [18] and also deep learning itself [26, 23, 12] . It causes inherent numerical instability for optimization algorithms. Moreover,

37th Conference on Neural Information Processing Systems (NeurIPS 2023).

the answer to this question depends on the architecture of the neural networks (specified by the number of layers, layers width, activation function, and so forth). A response to this question might suggest a guideline for model and architecture selection.

2. Theoretical viewpoint: the existence of optimal solutions is crucial for the analysis of algorithms and their properties (for example, the properties of convergence, or the characterization of properties of the optimum, related to the notion of implicit bias).

One usual practical (and also theoretical) trick to bypass the question of the existence of optimal solutions is to add a regularization term, which is usually coercive, e.g., the $L^2$ norm of the parameters. The existence of optimal solutions then follows by a classical argument on the extrema of a continuous function in a compact domain. Nevertheless, there are many settings where minimizing the regularized version might result in a high value of the loss since the algorithm has to make a trade-off between the loss and the regularizer. Such a scenario is discussed in Example 3.1. Therefore, studying the existence of optimal solutions without (explicit) regularization is also a question of interest.

Given a training set $\{(x_i, y_i)\}_{i=1}^{P}$, the problem of the existence of optimal solutions can be studied from the point of view of the set of functions implementable by the considered network architecture on the finite input domain $\Omega = \{x_i\}_{i=1}^{P}$. This is the case since the loss is usually of the form $\ell(f_\theta(x_i), y_i)$ where $f_\theta$ is the realization of the neural network with parameters $\theta$. Therefore, the loss involves directly the image of $\{x_i\}_{i=1}^{P}$ under the function $f_\theta$. For theoretical purposes, we also study the function space on the domain $\Omega = [-B, B]^d, B > 0$. In particular, we investigate two topological properties of these function spaces, both w.r.t. the infinity norm $\| \cdot \|_\infty$: the best approximation property (BAP), i.e., the guaranteed existence of an optimal solution $\theta^*$, and the closedness, a necessary property for the BAP. These properties are studied in Section 3 and Section 4, respectively. Most of our analysis is dedicated to the case of *regression problems*. We do make some links to the case of classification problems in Section 3.

We particularly focus on analyzing the function space associated with *(structured) sparse ReLU neural networks*, which is motivated by recent advances in machine learning witnessing a compelling empirical success of sparsity based methods in NNs and deep learning techniques, such as pruning [32, 14], sparse designed NN [4, 5], or the lottery ticket hypothesis [10] to name a few. Our approach exploits the notion of networks *either with fixed sparsity level or with fixed sparsity pattern (or support)*. This allows us to establish results covering both classical NNs (whose weights are not contrained to be sparse) and sparse NNs architectures. Our main contributions are:

1. **To study the BAP (i.e., the existence of optimal solutions) in practical problems (finite $\Omega$):** we provide a necessary condition and a sufficient one on the architecture (embodied by a sparsity pattern) to guarantee such existence. As a particular consequence of our results, we show that: a) *for one-hidden-layer NNs with a fixed sparsity level*, the training problem on a finite data set *always admits an optimal solution* (cf. Theorem 3.4 and Corollary 3.1); b) however, practitioners should be cautious since *there also exist fixed sparsity patterns that do not guarantee the existence of optimal solutions* (cf. Theorem 3.1 and Example 3.1). In the context of an emerging emphasis on *structured sparsity* (e.g. for GPU-friendliness), this highlights the importance of choosing adequate sparsity patterns.

2. **To study the closedness of the function space on $\Omega = [-B, B]^d$.** As in the finite case, we provide a necessary condition and a sufficient one for the closedness of the function space of ReLU NNs with a fixed sparsity pattern. In particular, our sufficient condition on one-hidden-layer networks generalizes the closedness results of [26, Theorem 3.8] on "dense' one-hidden-layer ReLU NNs to the case of sparse ones, either with fixed sparsity pattern (cf. Theorem 4.2, Corollary 4.1 and Corollary 4.2) or fixed sparsity level (Corollary 4.3). Moreover, our necessary condition (Theorem 4.1), which is also applicable to deep architectures, exhibits sparsity structures failing the closedness property.

Table 1 and Table 2 summarize our results and their positioning with respect to existing ones. Somewhat surprisingly, the necessary conditions in both domains ($\Omega$ finite and $\Omega = [-B, B]^d$) are identical. Our necessary/sufficient conditions also suggest a relation between sparse ReLU neural networks and their linear counterparts.

The rest of this paper is organized as follows: Section 2 discusses related works and introduces notations; the two technical sections, Section 3 and Section 4, presents the results for the case $\Omega$ finite set and $\Omega = [-B, B]^d$ respectively.

| Works | Architecture | Activation functions | $\Omega$ | Function space | BAP |
|---|---|---|---|---|---|
| Theorem 3.4 Corollary 3.1 | Sparse feed-forward network | ReLU | finite set | $(\mathbb{R}^{P \times d_o}, \|\cdot\|)$, arbitrary $\|\cdot\|$ | ✓ |
| [16][15] | Feed-forward network | Heavyside | $[0,1]^d$ | $(L^p(\Omega), \|\cdot\|_{L^p}), \forall p \in [1,\infty)$ | ✓ |
| [9][8]◇ | Feed-forward network, Residual feed-forward network | ReLU | $\mathbb{R}^d$ | $(L^p_\mu(\Omega), \|\cdot\|_{L^p}), p=2, \mu$ is a measure with compact support and is continuous w.r.t Lebesgue measure | ✓ (if the target function is continuous) |
| [26] | Feedforward network | ReLU, pReLU | $[-B,B]^d$ | $(C^0(\Omega), \|\cdot\|_\infty)$ | ✗ |
| Corollary 4.1† | Feed-forward network | ReLU | $[-B,B]^d$ | $(C^0(\Omega), \|\cdot\|_\infty)$ | ✗ |
| Corollary 4.2 Corollary 4.3 | Sparse feed-forward network | ReLU | $[-B,B]^d$ | $(C^0(\Omega), \|\cdot\|_\infty)$ | ✗ |

Table 1: **Closedness** results. All results are established for *one-hidden-layer* architectures with *scalar-valued* output, except † (which is valid for one-hidden-layer architectures with vector-valued output). In ◇, if the architecture is simply a feed-forward network, then the result is valid for any $p > 1$.

| Works | Architecture | Activation functions | Function space | Assumptions are valid for any … | | |
|---|---|---|---|---|---|---|
| | | | | $L$ | $N_{L-1}$ | $N_L$ |
| [12] | Feedforward network | Sigmoid | $(C^0(\Omega), \|\cdot\|_\infty)$ | ✗ $(L=2)$ | ✗ $(N_{L-1} \geq 2)$ | ✗ $(N_L=1)$ |
| [20]◇ | Feedforward network | ReLU | $(\mathbb{R}^{N_L \times P}, \|\cdot\|)$, $P=6$ | ✗ $(L=2)$ | ✗ $(N_{L-1}=2)$ | ✗ $(N_L=2)$ |
| [26] | Feedforward network | sigmoid, tanh, arctan, ISRLU, ISRU | $(C^0(\Omega), \|\cdot\|_\infty)$ | ✓ | ✗ $(N_{L-1} \geq 2)$ | ✗ $(N_L=1)$ |
| | | sigmoid, tanh, arctan, ISRLU, ISRU, ReLU, pReLU | $(L^p(\Omega), \|\cdot\|_{L^p})$ | | | |
| [23] | Feedforward network | ELU, softsign | $(W^{1,p}(\Omega), \|\cdot\|_{L^p})$ $\forall p \in [1,\infty]$ | | | |
| | | ISRLU | $(W^{2,p}(\Omega), \|\cdot\|_{L^p})$ $\forall p \in [1,\infty]$ | | | |
| | | ISRU, sigmoid, tanh, arctan | $(W^{k,p}(\Omega), \|\cdot\|_{L^p})$ $\forall k, \forall p \in [1,\infty]$ | | | |
| Theorem 4.1‡ | Sparse feedforward network | ReLU | $(C^0(\Omega), \|\cdot\|_\infty)$ | ✓ | ✓ | ✓ |
| Theorem 3.1◇ | Sparse feedforward network | ReLU | $(\mathbb{R}^{N_L \times P}, \|\cdot\|)$ | ✓ | ✓ | ✓ |

Table 2: **Non-closedness** results (notations in Section 2). Previous results consider $\Omega = [-B,B]^d$; ours cover: ◇ a finite $\Omega$ with $P$ points; ‡ a bounded $\Omega$ with non-empty interior (this includes $\Omega = [-B,B]^d$).

## 2 Related works

The fact that optimal solutions may not exist in tensor decomposition problems is well-documented [7]. The cause of this phenomenon (also referred to as *ill-posedness* [7]) is the non-closedness of the set of tensors of order at least three and of rank at least two. Similar phenomena were shown to happen in various settings such as matrix completion [11, Example 2], robust principal component analysis [30] and sparse matrix factorization [18, Remark A.1]. Our work indeed establishes bridges between the phenomenon on sparse matrix factorization [18] and on sparse ReLU NNs.

There is also an active line of research on the best approximation property and closedness of function spaces of neural networks. Existing results can be classified into two categories: *negative* results, which demonstrate the non-closedness and *positive* results for those showing the closedness or best approximation property of function spaces of NNs. Negative results can notably be found in

102 [12, 26, 23], showing that the set of functions implemented as conventional multilayer perceptrons
103 with various activation functions such as Inverse Square Root Linear Unit (ISRLU), Inverse Square
104 Root Unit (ISRU), parametric ReLU (pReLU), Exponential Linear Unit (ELU) [26, Table 1] is not
105 a closed subset of classical function spaces (e.g., the Lebesgue spaces $L^p$, the set of continuous
106 functions $C^0$ equipped with the sup-norm, or Sobolev spaces $W^{k,p}$). In a more practical setting, [20]
107 hand-crafts a dataset of six points which makes the training problem of a dense one-hidden-layer
108 neural network not admit any solution. Positive results are proved in [26, 16, 15], which establish
109 both the closedness and/or the BAP. The BAP implies closedness [12, Proposition 3.1][26, Section
110 3] (but the converse is not true, see Appendix D) hence the BAP can be more difficult to prove
111 than closedness. So far, the only architecture proved to admit the best approximation property (and
112 thus, also closedness) is *one-hidden-layer neural networks* with *heavyside activation* function and
113 *scalar-valued output* (i.e., output dimension equal to *one*) [15] in $L^p(\Omega), \forall p \in [1, \infty]$. If one allows
114 additional assumptions such as the target function $f$ being continuous, then BAP is also established
115 for one-hidden layer and residual one-hidden-layer NNs with ReLU activation function [9, 8]. In all
116 other settings, to the best of our knowledge, the only property proved in the literature is closedness,
117 but the BAP remains elusive. We compare our results with existing works in Tables 1 and 2.

118 In machine learning, there is an ongoing endeavour to explore sparse deep neural networks, as a
119 prominent approach to reduce memory and computation overheads inherent in deep learning. One
120 of its most well-known methods is Iterative Magnitude Pruning (IMP), which iteratively trains and
121 prunes connections/neurons to achieve a certain level of sparsity. This method is employed in various
122 works [14, 32], and is related to the so-called Lottery Ticket Hypothesis (LTH) [10]. The main issue
123 of IMP is its running time: one typically needs to perform many steps of pruning and retraining
124 to achieve a good trade-off between sparsity and performance. To address this issue, many works
125 attempt to identify the sparsity patterns of the network before training. Once they are found, it is
126 sufficient to train the sparse neural networks once. These *pre-trained* sparsity patterns can be found
127 through algorithms [29, 31, 19] or leveraging the sparse structure of well-known fast linear operators
128 such as the Discrete Fourier Transform [5, 4, 21, 6, 3]. Regardless of the approaches, these methods
129 are bound to train a neural network with *fixed sparsity pattern* at some points. This is a particular
130 motivation for our work and our study on the best approximation property of sparse ReLU neural
131 networks with fixed sparsity pattern.

132 **Notations**   In this work, $[\![n]\!] := \{1, \ldots, n\}$. For a matrix $\mathbf{A} \in \mathbb{R}^{m \times n}$, $\mathbf{A}[i, j]$ denotes the coefficient
133 at the index $(i, j)$; for subsets $S_r \subseteq [\![m]\!], S_c \subseteq [\![n]\!]$, $\mathbf{A}[S_r, :]$ (resp. $\mathbf{A}[:, S_c]$) is a matrix of the same
134 size as $\mathbf{A}$ and agrees with $\mathbf{A}$ on rows in $S_r$ (resp. columns in $S_c$) of $\mathbf{A}$ while its remaining coefficients
135 are zero. The operator $\mathrm{supp}(\mathbf{A}) := \{(\ell, k) \mid \mathbf{A}[\ell, k] \neq 0\}$ returns the *support* of the matrix $\mathbf{A}$. We
136 denote $\mathbf{1}_{m \times n}$ (resp. $\mathbf{0}_{m \times n}$) an all-one (resp. all-zero) matrix of size $m \times n$.

137 An architecture with fixed sparsity pattern is specified via $\mathbf{I} = (I_L, \ldots, I_1)$, a collection of binary
138 masks $I_i \in \{0, 1\}^{N_i \times N_{i-1}}, 1 \leq i \leq L$, where the tuple $(N_L, \ldots, N_0)$ denotes the dimensions of
139 the input layer $N_0 = d$, hidden layers $(N_{L-1}, \ldots, N_1)$ and output layer $(N_L)$, respectively. The
140 binary mask $I_i$ encodes the support constraints on the $i$th weight matrix $\mathbf{W}_i$, i.e., $I_i[\ell, k] = 0$ implies
141 $\mathbf{W}_i[\ell, k] = 0$. It is also convenient to think of $I_i$ as the set $\{(\ell, k) \mid I_i[\ell, k] = 1\}$, a subset of
142 $[\![N_i]\!] \times [\![N_{i-1}]\!]$. We will use these two interpretations (binary mask and subset) interchangeably and
143 the meaning should be clear from context. We will even abuse notations by denoting $I_l \subseteq \mathbf{1}_{N_l \times N_{l-1}}$.
144 Because the support constraint $I$ can be thought as a binary matrix, the notation $I[S_r, :]$ (resp. $I[:, S_c]$)
145 represents the support constraint of $I \cap S_r \times [\![n]\!]$ (resp. $I \cap [\![n]\!] \times S_c$).

146 The space of parameters on the sparse architecture $\mathbf{I}$ is denoted $\mathcal{N}_\mathbf{I}$, and for each $\theta \in \mathcal{N}_\mathbf{I}$, $\mathcal{R}_\theta :$
147 $\mathbb{R}^{N_0} \mapsto \mathbb{R}^{N_L}$ is the function implemented by the ReLU network with parameter $\theta$:

$$\mathcal{R}_\theta : x \in \mathbb{R}^{N_0} \mapsto \mathcal{R}_\theta(x) := \mathbf{W}_L \sigma(\ldots \sigma(\mathbf{W}_1 x + \mathbf{b}_1) \ldots + \mathbf{b}_{L-1}) + \mathbf{b}_L \in \mathbb{R}^{N_L} \qquad (1)$$

148 where $\sigma(x) = \max(0, x)$ is the ReLU activation.

149 Finally, for a given architecture $\mathbf{I}$, we define

$$\mathcal{L}_\mathbf{I} = \{\mathbf{X}_L \ldots \mathbf{X}_1 \mid \mathrm{supp}(\mathbf{X}_i) \subseteq I_i, i \in [\![L]\!]\} \subseteq \mathbb{R}^{N_L \times N_0} \qquad (2)$$

150 the set of matrices factorized into $L$ factors respecting the support constraints $I_i, i \in [\![L]\!]$. In fact, $\mathcal{L}_\mathbf{I}$
151 is the set of linear operators implementable as *linear* neural networks (i.e., with $\sigma = \mathrm{id}$ instead of
152 the ReLU in (1), and no biases) with parameters $\theta \in \mathcal{N}_\mathbf{I}$.

## 3  Analysis of fixed support ReLU neural networks for finite $\Omega$

The setting of a finite set $\Omega = \{x_i\}_{i=1}^P$ is common in many practical machine learning tasks: models such as (sparse) neural networks are trained on often large (but finite) annotated dataset $\mathcal{D} = \{(x_i, y_i)\}_{i=1}^P$. The optimization/training problem usually takes the form:

$$\underset{\theta}{\text{Minimize}} \qquad \mathcal{L}(\theta) = \sum_{i=1}^P \ell(\mathcal{R}_\theta(x_i), y_i), \qquad \text{under sparsity constraints on } \theta \qquad (3)$$

where $\ell$ is a loss function measuring the similarity between $\mathcal{R}_\theta(x_i)$ and $y_i$. A natural question that we would like to address for this task is:

**Question 3.1.** *Under which conditions on* **I***, the prescribed sparsity pattern for $\theta$, does the training problem of sparse neural networks admit an optimal solution for any finite data set $\mathcal{D}$?*

We investigate this question both for parameters $\theta$ constrained to satisfy a *fixed* sparsity pattern **I**, and in the case of a fixed sparsity level, see e.g. Corollary 4.3.

After showing in Section 3.1 that the answer to Question 3.1 is intimately connected with the closedness of the function space of neural networks with architecture **I**, we establish in Section 3.2 that this closedness implies the closedness of the matrix set $\mathcal{L}_{\mathbf{I}}$ (a property that can be checked using algorithms from real algebraic geometry, see Section 3.3). We also provide concrete examples of support patterns **I** where closedness provably fails, and neural network training can diverge. Section 3.4 presents sufficient conditions for closedness that enable us to show that an optimal solution always exists on scalar-valued one-hidden-layer networks under a constraint on the sparsity level of each layer.

### 3.1  Equivalence between closedness and best approximation property

To answer Question 3.1, it is convenient to view $\Omega$ as the matrix $[x_1, \ldots, x_P] \in \mathbb{R}^{d \times P}$ and to consider the function space implemented by neural networks with the given architecture **I** on the input domain $\Omega$ in dimension $d = N_0$, with output dimension $N_L$, defined as the set

$$\mathcal{F}_{\mathbf{I}}(\Omega) := \{\mathcal{R}_\theta(\Omega) \mid \theta \in \mathcal{N}_{\mathbf{I}}\} \subseteq \mathbb{R}^{N_L \times P} \qquad (4)$$

where the matrix $\mathcal{R}_\theta(\Omega) := [\mathcal{R}_\theta(x_1), \ldots, \mathcal{R}_\theta(x_P)] \in \mathbb{R}^{N_L \times P}$ is the image under $\mathcal{R}_\theta$ of $\Omega$.

We study the closedness of $\mathcal{F}_{\mathbf{I}}(\Omega)$ under the usual topology induced by any norm $\|\cdot\|$ of $\mathbb{R}^{N_L \times P}$. This property is interesting because if $\mathcal{F}_{\mathbf{I}}(\Omega)$ is closed for any $\Omega = \{x_i\}_{i=1}^P$, then an optimal solution is guaranteed to exist for any $\mathcal{D}$ under classical assumptions of $\ell(\cdot, \cdot)$. The following result is not difficult to prove, we nevertheless provide a proof in Appendix B.1 for completeness.

**Proposition 3.1.** *Assume that, for any fixed $y \in \mathbb{R}^{N_L}$, $\ell(\cdot, y) : \mathbb{R}^{N_L} \mapsto \mathbb{R}$ is continuous, coercive and that $y = \arg\min_{y'} \ell(y', y)$. For any sparsity pattern* **I** *with input dimension $N_0 = d$ the following properties are equivalent:*

*1. irrespective of the training set, problem (3) under the constraint $\theta \in \mathcal{N}_{\mathbf{I}}$ has an optimal solution;*

*2. for every $P$ and every $\Omega \in \mathbb{R}^{d \times P}$, the function space $\mathcal{F}_{\mathbf{I}}(\Omega)$ is a closed subspace of $\mathbb{R}^{N_L \times P}$.*

The assumption on $\ell$ is natural and realistic in *regression* problems: any loss function based on any norm on $\mathbb{R}^d$ (e.g. $\ell(y', y) = \|y' - y\|$), such as the quadratic loss, satisfies this assumption. In the classification case, using the soft-max after the last layer together with the cross-entropy loss function indeed leads to an optimization problem with no optimum (regardless of the architecture) when given a *single* training pair. This is due to the fact that changing either the bias or the scales of the last layer can lead the output of the soft-max arbitrarily close to an ideal Dirac mass. It is an interesting challenge to identify whether sufficiently many and diverse training samples (as in concrete learning scenarios) make the problem better posed, and amenable to a relevant closedness analysis.

In light of Proposition 3.1 we investigate next the closedness of $\mathcal{F}_{\mathbf{I}}(\Omega)$ for finite $\Omega$.

### 3.2  A necessary closedness condition for fixed support ReLU networks

Our next result reveals connections between the closedness of $\mathcal{F}_{\mathbf{I}}(\Omega)$ for finite $\Omega$ and the closedness of $\mathcal{L}_{\mathbf{I}}$, the space of sparse matrix products with sparsity pattern **I**.

**Theorem 3.1.** *If $\mathcal{F}_{\mathbf{I}}(\Omega)$ is closed for every finite $\Omega$ then $\mathcal{L}_{\mathbf{I}}$ is closed.*

Theorem 3.1 is a direct consequence of (and in fact logically equivalent to) the following lemma:

**Lemma 3.2.** *If $\mathcal{L}_{\mathbf{I}}$ is not closed then there exists a set $\Omega \subset \mathbb{R}^d$, $d = N_0$, of cardinality at most $P := (3N_0 4^{\sum_{i=1}^{L-1} N_i} + 1)^{N_0}$ such that $\mathcal{F}_{\mathbf{I}}(\Omega)$ is not closed.*

*Sketch of the proof.* Since $\mathcal{L}_{\mathbf{I}}$ is not closed, there exists $\mathbf{A} \in \overline{\mathcal{L}_{\mathbf{I}}} \setminus \mathcal{L}_{\mathbf{I}}$ ($\overline{\mathcal{L}}$ is the closure of the set $\mathcal{L}$). Considering $f(x) := \mathbf{A}x$, we construct a set $\Omega = \{x_i\}_{i=1}^P$ such that $[f(x_1), \dots, f(x_P)] \in \overline{\mathcal{F}_{\mathbf{I}}(\Omega)} \setminus \mathcal{F}_{\mathbf{I}}(\Omega)$. Therefore, $\mathcal{F}_{\mathbf{I}}(\Omega)$ is not closed. $\qquad\square$

The proof is in Appendix B.2. Besides showing a topological connection between $\mathcal{F}_{\mathbf{I}}$ (NNs with ReLU activation) and $\mathcal{L}_{\mathbf{I}}$ (linear NNs), Theorem 3.1 leads to a simple example where $\mathcal{F}_{\mathbf{I}}$ is not closed.

**Example 3.1** (**LU** architecture). *Consider $\mathbf{I} = (I_2, I_1) \in \{0,1\}^{d \times d} \times \{0,1\}^{d \times d}$ where $I_1 = \{(i,j) \mid 1 \le i \le j \le d\}$ and $I_2 = \{(i,j) \mid 1 \le j \le i \le d\}$. Any pair of matrices $\mathbf{X}_2, \mathbf{X}_1 \in \mathbb{R}^{d \times d}$ such that $\mathrm{supp}(\mathbf{X}_i) \subseteq I_i, i = 1, 2$ are respectively lower and upper triangular matrices. Therefore, $\mathcal{L}_{\mathbf{I}}$ is the set of matrices that admit an* exact *lower - upper (LU) factorization/decomposition. That explains its name: **LU** architecture. This set is well known to a) contain an open and dense subset of $\mathbb{R}^{d \times d}$; b) be strictly contained in $\mathbb{R}^{d \times d}$ [13, Theorem 3.2.1] [25, Theorem 1]. Therefore, $\mathcal{L}_{\mathbf{I}}$ is not closed and by the contraposition of Theorem 3.1 we conclude that there exists a finite set $\Omega$ such that $\mathcal{F}_{\mathbf{I}}(\Omega)$ is not closed.*

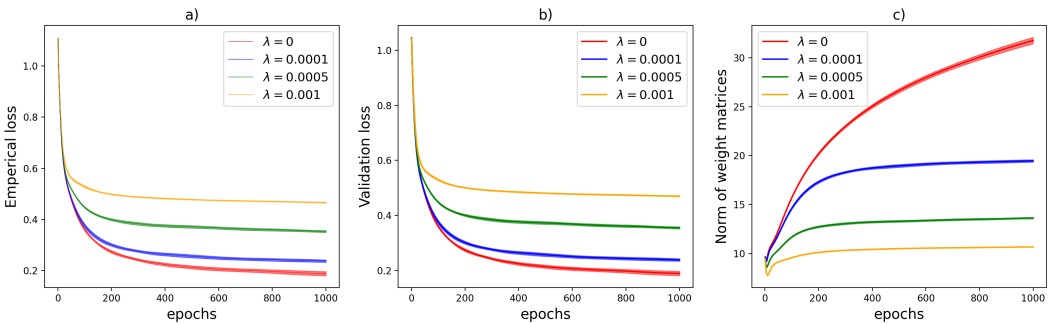

Figure 1: Training a one-hidden-layer fixed support (LU architecture) neural network with different regularization hyperparameters $\lambda$ (we use weight decay, i.e., an $L^2$ regularizer). Subfigures a)-b) show the relative loss (the lower, the better) for training (empirical loss) and testing (validation loss) respectively. Subfigure c) shows the norm of two weight matrices. The experiments are conducted 10 times to produce the error bars in all figures (almost invisible due to a small variability).

Let us illustrate the impact of the non-closedness in Example 3.1 via the behavior during the training of a fixed support one-hidden-layer neural network with the LU support constraint $\mathbf{I}$. This network is trained to learn the linear function $f(x) := \mathbf{A}x$ where $\mathbf{A} \in \mathbb{R}^{d \times d}$ is an anti-diagonal *identity* matrix. Using the necessary and sufficient condition of **LU** decomposition existence [25, Theorem 1], we have that $\mathbf{A} \in \overline{\mathcal{L}_{\mathbf{I}}} \setminus \mathcal{L}_{\mathbf{I}}$ as in the sketch proof of Lemma 3.2. Given network parameters $\theta$ and a training set, approximation quality can be measured by the relative loss: $\frac{1}{P}(\sum_{i=1}^P \|\mathcal{R}_\theta(x_i) - y_i\|_2^2 / \|y_i\|_2^2)$.

Figure 1 illustrates the behavior of the relative errors of the training set, validation set and the sum of weight matrices norm along epochs, using Stochastic Gradient Descent (SGD) with batch size 3000, learning rate 0.1, momentum 0.9 and four different weight decays (the hyperparameter controlling the $L^2$ regularizer) $\lambda \in \times\{0, 10^{-4}, 5 \times 10^{-4}, 10^{-3}\}$. The case $\lambda = 0$ corresponds to the *unregularized* case. Our training and testing sets contain each $P = 10^5$ samples generated independently as $x_i \sim \mathcal{U}([-1,1]^d)$ ($d = 100$) and $y_i := \mathbf{A}x_i$.

Example 3.1 and Figure 1 also lead to two interesting remarks: while the $L^2$ regularizer (weight decay) does prevent the parameter divergence phenomenon, the empirical loss is improved when using the non-regularized version. This is the situation where adding a regularization term might be detrimental, as stated earlier. More interestingly, the size of the dataset is $10^5$, which is much smaller than the theoretical $P$ in Lemma 3.2. It is thus interesting to see if we can reduce the theoretical value of $P$, which is currently exponential w.r.t. to the input dimension.

### 3.3 The closedness of $\mathcal{L}_\mathbf{I}$ is algorithmically decidable

Theorem 3.1 leads to a natural question: given $\mathbf{I}$, how to check the closedness of $\mathcal{L}_\mathbf{I}$, a subset of $\mathbb{R}^{N_L \times N_0}$. To the best of our knowledge, there is not any study on the closedness of $\mathcal{L}_\mathbf{I}$ in the literature. It is, thus, not known whether deciding on the closedness of $\mathcal{L}_\mathbf{I}$ for a given $\mathbf{I}$ is polynomially tractable. In this work, we show it is at least decidable with a doubly-exponential algorithm. This algorithm is an application of *quantifier elimination*, an algorithm from real algebraic geometry [2].

**Lemma 3.3.** *Given* $\mathbf{I} = (I_L, \dots, I_1)$, *the closedness of* $\mathcal{L}_\mathbf{I}$ *is decidable with an algorithm of complexity* $O((4L)^{C^{k-1}})$ *where* $k = N_L N_0 + 1 + 2 \sum_{i=1}^{L} |L_i|$ *and* $C$ *is a universal constant.*

We prove Lemma 3.3 in Appendix B.4. Since the knowledge of $\mathbf{I}$ is usually available (either fixed before training [19, 31, 4, 21, 3] or discovered by a procedure before re-training [10, 14, 32]), the algorithm in Lemma 3.3 is able to verify whether the training problem might not admit an optimum. While such a doubly exponential algorithm in Lemma 3.3 is seemingly impractical in practice, small toy examples (for example, Example 3.1 with $d = 2$) can be verified using Z3Prover[1], a software implementing exactly the algorithm in Lemma 3.3. However, Z3Prover is already unable to terminate when run on the **LU** architecture of Example 3.1 with $d = 3$. This calls for more efficient algorithms to determine the closedness of $\mathcal{L}_\mathbf{I}$ given $\mathbf{I}$. The same algorithmic question can be also asked for $\mathcal{F}_\mathbf{I}$. We leave these problems (in this general form) as open questions.

In fact, if such a polynomial algorithm (to decide the closedness of $\mathcal{L}_\mathbf{I}$) exists, it can be used to answer the following interesting question:

**Question 3.2.** *If the supports of the weight matrices are randomly sampled from a distribution, what is the probability that the corresponding training problem potentially admits no optimal solutions?*

While simple, this setting does happen in practice since random supports/binary masks are considered a strong and common baseline for sparse DNNs training [22]. Thanks to Theorem 3.1, if $\mathcal{L}_\mathbf{I}$ is not closed then the support is "bad". Thus, to have an estimation of a *lower bound* on the probability of "bad" supports, we could sample the supports from the given distribution and use the polynomial algorithm in question to *decide* if $\mathcal{L}_\mathbf{I}$ is closed. Unfortunately, the algorithm in Lemma 3.3 has doubly exponential complexity, thus hindering its practical use. However, for one-hidden-layer NNs, there is a *polynomial* algorithm to *detect* non-closedness: intuitively, if the support constraint is "locally similar" to the **LU** structure, then $\mathcal{L}_\mathbf{I}$ is not closed. This result is elaborated in Appendix B.5 and Lemma B.8. The resulting detection algorithm can have false negatives (i.e., it can fail to detect more complex configurations where $\mathcal{L}_\mathbf{I}$ is not closed) but no false positive.

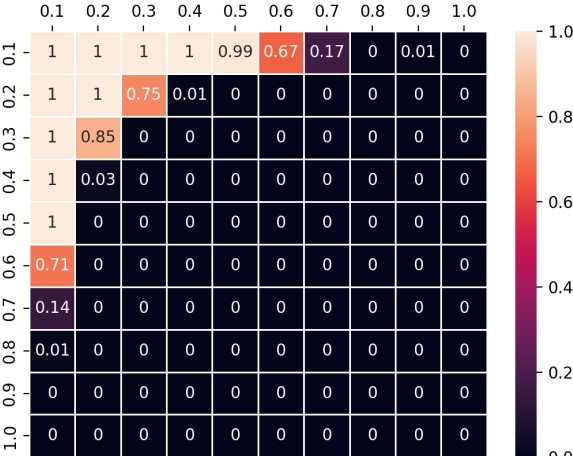

Figure 2: Probability of *detectable* "bad" support constraints sampled from uniform distribution over 100 samples.

---

[1]The package is developed by Microsoft research and it can be found at https://github.com/Z3Prover/z3

We test this algorithm on a one-hidden layer ReLU network with two $100 \times 100$ weight matrices. We randomly choose their supports whose cardinality are $p_1 \cdot 100^2$ and $p_2 \cdot 100^2$ respectively, with $(p_1, p_2) \in \{0.1, 0.2, \ldots, 1.0\}^2$. For each pair $(p_1, p_2)$, we sample 100 instances. Using the detection algorithm, we obtain Figure 2. The numbers in Figure 2 indicate the probability that a random support constraint $(I, J)$ has $\mathcal{E}_{I,J}$ non-closed (as detected by the algorithm). This figure shows two things: 1) "Bad" architectures such as **LU** are not rare and one can (randomly) generate plenty of them. 2) At a sparse regime ($a_1, a_2 \leq 0.2$), most of the random supports might lead to training problems without optimal solutions. We remind that the detection algorithm may give some false negatives. Thus, for less sparse regimes, it is possible that our heuristic fails to detect the non-closedness. The algorithm indeed gives a lower bound on the probability of finding non-closed instances. The code for Example 3.1, Question 3.2 and the algorithm in Lemma 3.3 is provided in [17].

### 3.4 Best approximation property of scalar-valued one-hidden-layer sparse networks

So far, we introduced a necessary condition for the closedness (and thus, by Proposition 3.1, the best approximation property) of sparse ReLU networks, and we provided an example of an architecture **I** whose training problem might not admit any optimal solution. One might wonder if there are architectures **I** that *avoid* the issue caused by the non-closedness of $\mathcal{F}_{\mathbf{I}}$. Indeed, we show that for one-hidden-layer sparse ReLU neural networks with scalar output dimension (i.e., $L = 2, N_2 = 1$), the existence of optimal solutions is guaranteed, *regardless of the sparsity pattern*.

**Theorem 3.4.** *Consider scalar-valued, one-hidden-layer ReLU neural networks (i.e., $L = 2, N_2 = 1$). For any support pairs $\mathbf{I} = (I_2, I_1)$ and any finite set $\Omega := \{x_1, \ldots, x_P\}$, $\mathcal{F}_{\mathbf{I}}(\Omega)$ is closed.*

The proof of Theorem 3.4 is deferred to Appendix B.3. As a sanity check, observe that when $L = 2, N_2 = 1$, the necessary condition in Theorem 3.1 is satisfied. Indeed, since $N_2 = 1$, $\mathcal{L}_{\mathbf{I}} \subseteq \mathbb{R}^{1 \times N_0}$ can be thought as a subset of $\mathbb{R}^{N_0}$. Any $\mathbf{X} \in \mathcal{L}_{\mathbf{I}}$ can be written as a sum: $\mathbf{X} = \sum_{i \in I_2} \mathbf{W}_2[i] \mathbf{W}_1[i, :]$, a decomposition of the product $\mathbf{W}_2 \mathbf{W}_1$, where $\mathbf{W}_2[i] \in \mathbb{R}, \mathbf{W}_1[i, :] \in \mathbb{R}^{N_0}$, $\text{supp}(\mathbf{W}_1[i, :]) \subseteq I_1[i, :]$. Define $\mathcal{H} := \cup_{i \in I_2} I_1[i, :] \subseteq [\![N_0]\!]$ the union of row supports of the first weight matrix. It is easy to verify that $\mathcal{L}_{\mathbf{I}}$ is isomorphic to $\mathbb{R}^{|\mathcal{H}|}$, which is closed. In fact, this argument only works for scalar-valued output, $N_2 = 1$. Thus, there is no conflict between Theorem 3.1 and Theorem 3.4.

In practice, many approaches search for the best support **I** *among a collection of possible supports*, for example, the approach of pruning and training [14, 32] or the lottery ticket hypothesis [10]. Our result for fixed support in Theorem 3.4 can be also applied in this case and is stated in Corollary 3.1. In particular, we consider a set of supports such that the support sizes (or sparsity ratios) of the layers are kept below a certain threshold $K_i, i = 1, \ldots, L$. This constraint on the sparsity level of each layer is widely used in many works on sparse neural networks [14, 32, 10].

**Corollary 3.1.** *Consider scalar-valued, one-hidden-layer ReLU neural networks. For any finite data set[2] $\mathcal{D} = (x_i, y_i)_{i=1}^P$, problem (3) under the constraints $\|\mathbf{W}_i\|_0 \leq K_i, i = 1, 2$ has a minimizer.*

*Proof.* Denote $\mathcal{I}$ the collection of sparsity patterns satisfying $\|I_i\|_0 \leq K_i, i = 1, 2$, so that a set of parameters satisfies the sparsity constraints $\|\mathbf{W}_i\|_0 \leq K_i, i = 1, 2$ if and only if the supports of the weight matrices belong to $\mathcal{I}$. Therefore, to solve the optimization problem under sparsity constraints $\|\mathbf{W}_i\|_0 \leq K_i, i = 1, 2$, it is sufficient to solve the same problem for every sparsity pattern in $\mathcal{I}$.

For each $\mathbf{I} \in \mathcal{I}$, we solve a training problem with architecture **I** on a given finite dataset $\mathcal{D}$. Thanks to Theorem 3.4 and Proposition 3.1, the infimum is attained. We take the optimal solution corresponding to **I** that yields the smallest value of the loss function $\mathcal{L}$. This is possible because the set $\mathcal{I}$ has a finite number of elements (the total number of possible sparsity patterns is finite). $\qquad \square$

## 4 Analysis of fixed support ReLU networks on continuous domains

We now investigate closedness properties when the domain $\Omega \subseteq \mathbb{R}^d$ is no longer finite. Denoting $\mathcal{F}_{\mathbf{I}} = \{\mathcal{R}_\theta : \mathbb{R}^{N_0} \mapsto \mathbb{R}^{N_L} \mid \theta \in \mathcal{N}_{\mathbf{I}}\}$ (with $N_0 = d$) the functions that can be implemented on a given ReLU network architecture **I**, we are interested in $\mathcal{F}_{\mathbf{I}}(\Omega) = \{f_{|\Omega} : f \in \mathcal{F}_{\mathbf{I}}\}$, the restriction of elements of $\mathcal{F}_{\mathbf{I}}$ to $\Omega$. This is a natural extension of the set $\mathcal{F}_{\mathbf{I}}(\Omega)$ studied in the case of finite $\Omega$.

---

[2] Notice that $\mathcal{D}$ contains both input vectors $x_i$ and targets $y_i$, unlike $\Omega$ which only contains the inputs.

Specifically, we investigate the closedness of $\mathcal{F}_{\mathbf{I}}(\Omega)$ in $(C^0(\Omega), \|\cdot\|_\infty)$ (the set of continuous functions on $\Omega$ equipped with the supremum norm $\|f\|_\infty := \sup_{x \in \Omega} \|f(x)\|_2$). Contrary to the previous section, we can no longer exploit Proposition 3.1 to deduce that the closedness property and the BAP are equivalent. The results in this section can be seen as a continuation (and also generalization) of the line of research on the topological property of function space of neural networks [26, 12, 16, 15, 23]. In Section 4.1 and Section 4.2, we provide a necessary and a sufficient condition on $\mathbf{I}$ for the closedness of $\mathcal{F}_{\mathbf{I}}(\Omega)$ in $(C^0(\Omega), \|\cdot\|_\infty)$ respectively. The condition of the former is valid for any depth, while that of the latter is applicable for one-hidden-layer networks $(L = 2)$. These results are established under various assumptions on $\Omega$ (such as $\Omega = [-B, B]^d$, or $\Omega$ being bounded with non-empty interior) that will be specified in each result.

## 4.1 A necessary condition for closedness of fixed support ReLU network

Theorem 4.1 states our result on the necessary condition for the closedness. Interestingly, observe that this result (which is proved in Appendix C.1) naturally generalizes Theorem 3.1. Again, closedness of $\mathcal{L}_{\mathbf{I}}$ in $\mathbb{R}^{N_L \times N_0}$ is with respect to the usual topology defined by any norm.

**Theorem 4.1.** *Consider $\Omega \subset \mathbb{R}^d$ a bounded set with non-empty interior, and $\mathbf{I}$ a sparse architecture with input dimension $N_0 = d$. If $\mathcal{F}_{\mathbf{I}}(\Omega)$ is closed in $(C^0(\Omega), \|\cdot\|_\infty)$ then $\mathcal{L}_{\mathbf{I}}$ is closed in $\mathbb{R}^{N_L \times N_0}$.*

Theorem 4.1 applies for any $\Omega$ which is bounded and has non-empty interior. Thus, it encompasses not only the hypercubes $[-B, B]^d, B > 0$ but also many other domains such as closed or open $\mathbb{R}^d$ balls. Similar to Theorem 3.1, Theorem 4.1 is interesting in the sense that it allows us to check the non-closedness of the function space $\mathcal{F}_{\mathbf{I}}$ (a subset of the infinite-dimensional space $C^0(\Omega)$) by checking that of $\mathcal{L}_{\mathbf{I}} \subseteq \mathbb{R}^{N_L \times N_0}$ (a finite-dimensional space). The latter can be checked using the algorithm presented in Lemma 3.3. Moreover, the **LU** architecture presented in Example 3.1 is also an example of $\mathbf{I}$ whose function space is not closed in $(C^0(\Omega), \|\cdot\|_\infty)$.

## 4.2 A sufficient condition for closedness of fixed support ReLU network

The following theorem is the main result of this section. It provides a sufficient condition to verify the closedness of $\mathcal{F}_{\mathbf{I}}(\Omega)$ for $\Omega = [-B, B]^d, B > 0$ with one-hidden-layer sparse ReLU neural networks.

**Theorem 4.2.** *Consider $\Omega = [-B, B]^d$, $N_0 = d$ and a sparsity pattern $\mathbf{I} = (I_2, I_1)$ such that:*

*1. There is no support constraint for the weight matrix of the second layer, $\mathbf{W}_2$: $I_2 = \mathbf{1}_{N_2 \times N_1}$;*

*2. For each non-empty set of hidden neurons, $S \subseteq [\![N_1]\!]$, $\mathcal{L}_{\mathbf{I}_S}$ is closed in $\mathbb{R}^{N_2 \times N_1}$, where $\mathbf{I}_S := (I_2[:, S], I_1[S, :])$ is the support constraint restricted to the sub-network with hidden neurons in $S$.*

*Then the set $\mathcal{F}_{\mathbf{I}}(\Omega)$ is closed in $(C^0(\Omega), \|\cdot\|_\infty)$.*

Both conditions in Theorem 4.2 can be verified algorithmically: while the first one is trivial to check, the second one requires us to check the closedness of at most $2^{N_1}$ sets $\mathcal{L}_{\mathbf{I}_S}$ (because there are at most $2^{N_1}$ subsets of $[\![N_1]\!]$), which is still algorithmically possible (although perhaps practically intractable) with the algorithm of Lemma 3.3. Apart from its algorithmic aspect, we present two interesting corollaries of Theorem 4.2. The first one, Corollary 4.1, is about the closedness of the function space of fully connected (i.e., with no sparsity constraint) one-hidden-layer neural networks.

**Corollary 4.1** (Closedness of fully connected one-hidden-layer ReLU networks *of any output dimension*). *Given $\mathbf{I} = (\mathbf{1}_{N_2 \times N_1}, \mathbf{1}_{N_1 \times N_0})$, the set $\mathcal{F}_{\mathbf{I}}$ is closed in $(C^0([-B, B]^d), \|\cdot\|_\infty)$ where $d = N_0$.*

*Proof.* The result follows from Theorem 4.2 once we check if its assumptions hold. The first one is trivial. To check the second, observe that for every non-empty set of hidden neurons $S \subseteq [\![N_1]\!]$, the set $\mathcal{L}_{\mathbf{I}_S} \subseteq \mathbb{R}^{N_2 \times N_0}$ is simply the set of matrices of rank at most $|S|$, which is closed for any $S$. $\qquad\square$

Corollary 4.2 states the closedness of scalar-valued, one-hidden-layer sparse ReLU NNs. In a way, it can be seen as the analog of Theorem 3.4 for $\Omega = [-B, B]^d$.

**Corollary 4.2** (Closedness of *fixed support* one-hidden-layer ReLU networks with scalar output). *Given any input dimension $d = N_0 \geq 1$, any number of hidden neurons $N_1 \geq 1$, scalar output dimension $N_2 = 1$, and any prescribed supports $\mathbf{I} = (I_2, I_1)$, the set $\mathcal{F}_{\mathbf{I}}$ is closed in $(C^0([-B, B]^d), \|\cdot\|_\infty)$.*

*Sketch of the proof.* If there exists a hidden neuron $i \in [\![N_1]\!]$ such that $I_2[i] = 0$ (i.e., $i \notin I_2$: $i$ is not connected to the only output of the network), we have: $\mathcal{F}_{\mathbf{I}} = \mathcal{F}_{\mathbf{I}'}$ where $\mathbf{I}' = \mathbf{I}_S$, $S = [\![N_1]\!] \setminus \{i\}$. By repeating this process, we can assume without loss of generality that $I_2[i] = \mathbf{1}_{1 \times N_1}$. That is the first condition of Theorem 4.2.

Therefore, it is sufficient to verify the second condition of Theorem 4.2. Consider any non-empty set of hidden neurons $S \subseteq [\![N_1]\!]$, and define $\mathcal{H} := \cup_{i \in S} I[i,:] \subseteq [\![N_0]\!]$ the union of row supports of $I_1[S,:]$. It is easy to verify that $\mathcal{L}_{\mathbf{I}_S}$ is isomorphic to $\mathbb{R}^{|\mathcal{H}|}$, which is closed. The result follows by Theorem 4.2. For a more formal proof, readers can find an inductive one in Appendix C.3. $\qquad\square$

In fact, both Corollary 4.1 and Corollary 4.2 generalize [26, Theorem 3.8], which proves the closedness of $\mathcal{F}_{\mathbf{I}}([-B, B]^d)$ when $I_2 = \mathbf{1}_{1 \times N_1}, I_1 = \mathbf{1}_{N_1 \times N_0}$ (classical fully connected one-hidden-layer ReLU networks with output dimension equal to one).

To conclude, let us consider the analog to Corollary 3.1: we study the function space implementable as a sparse one-hidden-layer network with constraints on the *sparsity level* of each layer (i.e., $\|\mathbf{W}_i\|_0 \leq K_i, i = 1, 2$.

**Corollary 4.3.** *Consider scalar-valued, one-hidden-layer ReLU networks* ($L = 2, N_2 = 1, N_1, N_0$) *with $\ell^0$ constraints $\|\mathbf{W}_1\|_0 \leq K_1, \|\mathbf{W}_2\|_0 \leq K_2$ for some constants $K_1, K_2 \in \mathbb{N}$. The function space $\mathcal{F}([-B, B]^d)$ associated with this architecture is closed in* ($C^0([-B, B]^{N_0}), \|\cdot\|_\infty$).

*Proof.* Denote $\mathcal{I} := \{(I_2, I_1) \mid I_2 \subseteq [\![1]\!] \times [\![N_1]\!], I_1 \subseteq [\![N_1]\!] \times [\![N_0]\!], |I_1| \leq K_1, |I_2| \leq K_2\}$ the set of sparsity patterns respecting the $\ell^0$ constraints, so that $\mathcal{F}([-B, B]^d) = \bigcup_{\mathbf{I} \in \mathcal{I}} \mathcal{F}_{\mathbf{I}}([-B, B]^d)$. Since $\mathcal{I}$ is finite and $\forall \mathbf{I} \in \mathcal{I}, \mathcal{F}_{\mathbf{I}}([-B, B]^d)$ is closed (Corollary 4.2), the result is proved. $\qquad\square$

# 5 Conclusion

In this paper, we study the somewhat overlooked question of the existence of an optimal solution to sparse neural network training problems. The study is accomplished by adopting the point of view of topological properties of the function spaces of such networks on two types of domains: a finite domain $\Omega$, or (typically) a hypercube. On the one hand, our investigation of the BAP and the closedness of these function spaces reveals the existence of *pathological* sparsity patterns that fail to have optimal solutions on some instances (cf Theorem 3.1 and Theorem 4.1) and thus possibly cause instabilities in optimization algorithms (see Example 3.1 and Figure 1). On the other hand, we also prove several positive results on the BAP and closedness, notably for sparse one-hidden-layer ReLU neural networks (cf. Theorem 3.4 and Theorem 4.2). These results provide new instances of network architectures where the BAP is proved (cf Theorem 3.4) and substantially generalize existing ones (cf. Theorem 4.2).

In the future, a particular theoretical challenge is to propose necessary and sufficient conditions for the BAP and closedness of $\mathcal{F}_{\mathbf{I}}(\Omega)$, if possible covering in a single framework both types of domains $\Omega$ considered here. The fact that the conditions established on these two types of domains are very similar (cf. the similarity between Theorem 3.1 and Theorem 4.1, as well as between Theorem 3.4 and Corollary 4.2) is encouraging. Another interesting algorithmic challenge is to substantially reduce the complexity of the algorithm to decide the closedness of $\mathcal{L}_{\mathbf{I}}$ in Lemma 3.3, which is currently doubly exponential. It calls for a more efficient algorithm to make this check more practical. Achieving a practically tractable algorithm would for instance allow to check if a support selected e.g. by IMP is pathological or not. This would certainly consolidate the algorithmic robustness and theoretical foundations of pruning techniques to sparsity deep neural networks. From a more theoretical perspective, the existence of an optimum solution in the context of classical linear inverse problems has been widely used to analyze the desirable properties of certain cost functions, e.g. $\ell^1$ minimization for sparse recovery. Knowing that an optimal solution exists for a given sparse neural network training problem is thus likely to open the door to further fruitful insights.

## Acknowledgement

This project was supported by the AllegroAssai ANR project ANR-19-CHIA-0009. The authors thank the Blaise Pascal Center (CBP) for the computational means. It uses the SIDUS [27] solution developed by Emmanuel Quemener. Q-T. Le wants to personally thank M-L.Nguyen [3] for his enlightenment on the non-equivalence between BAP and closedness in infinite dimension space in Appendix D.

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

## A  Additional notations

In this work, matrices are written in bold uppercase letters. Vectors are written in bold lowercase letters only if they indicate network parameters (such as bias). For a matrix $\mathbf{A} \in \mathbb{R}^{m \times n}$, we use $\mathbf{A}[i,:] \in \mathbb{R}^{1 \times n}$ (resp. $\mathbf{A}[:,i] \in \mathbb{R}^{m \times 1}$) to denote the row (resp. column) vector corresponding the $i$th row (resp. column) of $\mathbf{A}$. To ease the notation, we write $\mathbf{A}[i,:]\mathbf{v}$ to denote the scalar product between $\mathbf{A}[i,:]$ and the vector $\mathbf{v} \in \mathbb{R}^n$. This notation will be used regularly when we decompose the functions of one-hidden-neural networks into sum of functions corresponding to hidden neurons.

For a vector $v \in \mathbb{R}^d$, $v[I] \in \mathbb{R}^{|I|}$ is the vector $v$ restricted to coefficients in $I \subseteq [\![d]\!]$. If $I = \{i\}$ a singleton, $v[i] \in \mathbb{R}$ is the $i$th coefficient of $v$. We also use $\mathbf{1}_m$ and $\mathbf{0}_m$ to denote an all-one (resp. all-zero) vector of size $m$.

For a *dense* (fully connected) feedforward architecture, we denote $\mathbf{N} = (N_L, \ldots, N_0)$ the dimensions of the input layer $N_0 = d$, hidden layers $(N_{L-1}, \ldots, N_1)$ and output layer $(N_L)$, respectively. The parameters space of the dense architecture $\mathbf{N}$ is denoted by $\mathcal{N}_\mathbf{N}$: it is the set of all coefficients of the weight matrices $\mathbf{W}_i \in \mathbb{R}^{N_i \times N_{i-1}}$ and bias vectors $\mathbf{b}_i \in \mathbb{R}^{N_i}, i = 1, \ldots, L$. It is easy to verify that $\mathcal{N}_\mathbf{N}$ is isomorphic to $\mathbb{R}^N$ where $N = \sum_{i=1}^L N_{i-1}N_i + \sum_{i=1}^L N_i$ is the total number of parameters of the architecture.

Clearly, $\mathcal{N}_\mathbf{I} \subseteq \mathcal{N}_\mathbf{N}$ since:

$$\mathcal{N}_\mathbf{I} := \{\theta = ((\mathbf{W}_i, \mathbf{b}_i))_{i=1,\ldots,L} : \mathrm{supp}(\mathbf{W}_i) \subseteq I_i, \forall i = 1, \ldots, L.\}. \tag{5}$$

A special subset of $\mathcal{N}_\mathbf{I}$ is the set of network parameters with zero biases,

$$\mathcal{N}_\mathbf{I}^0 := \{\theta = ((\mathbf{W}_i, \mathbf{0}_{N_i}))_{i=1,\ldots,L} : \mathrm{supp}(\mathbf{W}_i) \subseteq I_i, \forall i = 1, \ldots, L.\}. \tag{6}$$

Given an activation function $\nu$, the realization $\mathcal{R}_\theta^\nu$ of a neural network $\theta \in \mathcal{N}_\mathbf{N}$ is the function

$$\mathcal{R}_\theta^\nu : x \in \mathbb{R}^{N_0} \mapsto \mathcal{R}_\theta^\nu(x) := \mathbf{W}_L \nu(\ldots \nu(\mathbf{W}_1 x + \mathbf{b}_1) \ldots + \mathbf{b}_{L-1}) + \mathbf{b}_L \in \mathbb{R}^{N_L} \tag{7}$$

We denote $\mathcal{R}^\nu : \theta \mapsto \mathcal{R}_\theta^\nu$ the functional mapping from a set of parameters $\theta$ to its realization. The function space associated to a sparse architecture $\mathbf{I}$ and activation function $\nu$ is the image of $\mathcal{N}_\mathbf{I}$ under $\mathcal{R}^\nu$:

$$\mathcal{F}_\mathbf{I}^\nu := \mathcal{R}^\nu(\mathcal{N}_\mathbf{I}). \tag{8}$$

When $\nu = \sigma$ the ReLU activation function, we recover the definition of realization in Equation (1). We use the shorthands

$$\begin{aligned} \mathcal{R}_\theta &:= \mathcal{R}_\theta^\sigma \\ \mathcal{F}_\mathbf{I} &:= \mathcal{F}_\mathbf{I}^\sigma, \end{aligned} \tag{9}$$

as in the main text. This allows us to define $\mathcal{L}_\mathbf{I}$ (cf. Equation (2)) as $\mathcal{L}_\mathbf{I} := \mathcal{R}^{\mathrm{Id}}(\mathcal{N}_\mathbf{I}^0)$ where $\nu = \mathrm{Id}$ is the identity map, which is a subset of linear maps $\mathbb{R}^{N_0} \mapsto \mathbb{R}^{N_L}$.

## B  Proofs for results in Section 3

### B.1  Proof of Proposition 3.1

*Proof.* First, we remind the problem of the training of a sparse neural network on a finite data set $\mathcal{D} = \{(x_i, y_i)\}_{i=1}^P$:

$$\underset{\theta \in \mathcal{N}_\mathbf{I}}{\text{Minimize}} \quad \mathcal{L}(\theta) := \sum_{i=1}^P \ell(\mathcal{R}_\theta(x_i), y_i), \tag{10}$$

which shares the same optimal value as the following optimization problem:

$$\underset{\mathbf{D} \in \mathcal{F}_\mathbf{I}(\Omega)}{\text{Minimize}} \quad \mathcal{L}(\mathbf{D}) := \sum_{i=1}^P \ell(\mathbf{D}[:,i], y_i) \tag{11}$$

where $\Omega = \{x_i\}_{i=1}^P$. This is simply a change of variables: from $\mathcal{R}_\theta(x_i)$ to the $i$th column of $\mathbf{D} = \mathcal{R}_\theta(\Omega)$. We prove two implications as follows:

1. *Assume the closedness of $\mathcal{F}_{\mathbf{I}}(\Omega)$ for every finite $\Omega$. Then an optimal solution of the optimization problem* (10) *exists for every finite data set* $\{(x_i, y_i)\}_{i=1}^{P}$. Consider a training set $\{(x_i, y_i)\}_{i=1}^{P}$ and $\Omega := \{x_i\}_{i=1}^{P}$. Since $\mathbf{D} := \mathbf{0}_{P \times N_L} \in \mathcal{F}_{\mathbf{I}}(\Omega)$ (by setting all parameters in $\theta$ equal to zero), the set $\mathcal{F}_{\mathbf{I}}(\Omega)$ is non-empty. The optimal value of (11) is thus upper bounded by $\mathcal{L}(\mathbf{0})$. Since the function $\ell(\cdot, y_i)$ is coercive for every $y_i$ in the training set, there exists a constant $C$ (dependent on the training set and the loss) such that minimizing (11) on $\mathcal{F}_{\mathbf{I}}(\Omega)$ or on $\mathcal{F}_{\mathbf{I}}(\Omega) \cap \mathcal{B}(0, C)$ (with $\mathcal{B}(\mathbf{0}, C)$ the $L^2$ ball of radius $C$ centered at zero) yields the same infimum. The function $\mathcal{L}$ is continuous, since each $\ell(\cdot, y_i)$ is continuous by assumption, and the set $\mathcal{F}_{\mathbf{I}}(\Omega) \cap \mathcal{B}(0, C)$ is compact, since it is closed (as an intersection of two closed sets) and bounded (since $\mathcal{B}(0, C)$ is bounded). As a result there exists a matrix $\mathbf{D} \in \mathcal{F}_{\mathbf{I}}(\Omega) \cap \mathcal{B}(0, C)$ yielding the optimal value for (11). Thus, the parameters $\theta$ such that $\mathcal{R}_\theta(\Omega) = \mathbf{D}$ is an optimal solution of (10).

2. *Assume that an optimal solution of problem 10 exists for every finite data set* $\{(x_i, y_i)\}_{i=1}^{P}$. *Then* $\mathcal{F}_{\mathbf{I}}(\Omega)$ *is closed for every $\Omega$ finite.* We prove the contraposition of this claim. Assume there exists a finite set $\Omega = \{x_i\}_{i=1}^{P}$ such that $\mathcal{F}_{\mathbf{I}}(\Omega)$ is not closed. Then, there exists a matrix $\mathbf{D} \in \mathbb{R}^{N_L \times P}$ such that $\mathbf{D} \in \overline{\mathcal{F}_{\mathbf{I}}(\Omega)} \setminus \mathcal{F}_{\mathbf{I}}(\Omega)$. Consider the dataset $\{(x_i, y_i)\}_{i=1}^{P}$ where $y_i \in \mathbb{R}^{N_L}$ is the $i$th column of $\mathbf{D}$. We prove that the infimum value of (10) is $V := \sum_{i=1}^{P} \ell(y_i, y_i)$. Indeed, since $\mathbf{D} \in \overline{\mathcal{F}_{\mathbf{I}}(\Omega)}$, there exists a sequence $\{\theta_k\}_{k \in \mathbb{N}}$ such that $\lim_{k \to \infty} \mathcal{R}_{\theta_k}(\Omega) = \mathbf{D}$. Therefore, by continuity of $\ell(\cdot, y_i)$, we have:

$$\lim_{k \to \infty} \mathcal{L}(\theta_k) = \sum_{i=1}^{P} \lim_{k \to \infty} \ell(\mathcal{R}_{\theta_k}(x_i), y_i) = \sum_{i=1}^{P} \ell(y_i, y_i) = V.$$

Moreover, the infimum cannot be smaller than $V$ because the $i$th summand is at least $\ell(y_i, y_i)$ (due to the assumption on $\ell$ in Proposition 3.1). Therefore, the infimum value is indeed $V$. Since we assume that $y$ is the *only* minimizer of $y' \mapsto \ell(y', y)$, this value can be achieved only if there exists a parameter $\theta \in \mathbf{I}$ such that $\mathcal{R}_\theta(\Omega) = \mathbf{D}$. This is impossible due to our choice of $\mathbf{D}$ which *does not* belong to $\mathcal{F}_{\mathbf{I}}(\Omega)$. We conclude that with our constructed data set $\mathcal{D}$, an optimal solution *does not* exist for (10). □

## B.2 Proof of Lemma 3.2

The proof of Lemma 3.2 (and thus, as discussed in the main text, of Theorem 3.1) use four technical lemmas. Lemma B.1 is proved in Appendix C.1 since it involves Theorem 4.1. The other lemmas are proved right after the proof of Lemma 3.2.

**Lemma B.1.** *If $\mathbf{A} \in \overline{\mathcal{L}_{\mathbf{I}}} \setminus \mathcal{L}_{\mathbf{I}} \subseteq \mathbb{R}^{N_L \times N_0}$ then the function $f : x \mapsto f(x) := \mathbf{A}x$ satisfies $f \in \overline{\mathcal{F}_{\mathbf{I}}(\Omega)} \setminus \mathcal{F}_{\mathbf{I}}(\Omega)$ for every subset $\Omega$ of $\mathbb{R}^{N_0}$ that is bounded with non-empty interior.*

**Lemma B.2.** *Consider $\Omega = \{x_i\}_{i=1}^{P}$ a finite subset of $\mathbb{R}^d$ and $\Omega' = [-B, B]^d$ such that $\Omega \subseteq \Omega'$. If $f \in \overline{\mathcal{F}_{\mathbf{I}}(\Omega')}$ (under the topology induced by $\|\cdot\|_\infty$), then $\mathbf{D} := [f(x_1) \dots f(x_P)] \in \overline{\mathcal{F}_{\mathbf{I}}(\Omega)}$.*

**Lemma B.3.** *Consider $\mathcal{R}_\theta$, the realization of a ReLU neural network with parameter $\theta \in \mathbf{I}$. This function is continuous and piecewise linear. On the interior of each piece, its Jacobian matrix is constant and satisfies $\mathbf{J} \in \mathcal{L}_{\mathbf{I}}$.*

**Lemma B.4.** *For $p, N \in \mathbb{N}$, consider the following set of points (a discretized grid for $[0, 1]^N$):*

$$\Omega = \Omega_p^N = \left\{ \left( \frac{i_1}{p}, \dots, \frac{i_N}{p} \right) \mid 0 \le i_j \le p, i_j \in \mathbb{N}, \forall 1 \le j \le N \right\}.$$

*If $H \in \mathbb{N}$ satisfies $p \ge 3NH$, then for any collection of $H$ hyperplanes, there exists $x \in \Omega_p^N$ such that the elementary hypercube whose vertices are of the form*

$$\left\{ x + \left( \frac{i_1}{p}, \dots, \frac{i_N}{p} \right) \mid i_j \in \{0, 1\} \ \forall 1 \le j \le N \right\} \subseteq \Omega_p^N$$

*lies entirely inside a polytope delimited by these hyperplanes.*

We are now ready to prove Lemma 3.2.

*Proof of Lemma 3.2.* Since $\mathcal{L}_{\mathbf{I}}$ is not closed, there exists a matrix $\mathbf{A} \in \overline{\mathcal{L}_{\mathbf{I}}} \setminus \mathcal{L}_{\mathbf{I}}$, and we consider $f(x) := \mathbf{A}x$. Setting $p := 3N_0 4^{\sum_{i=1}^{L-1} N_i}$ we construct $\Omega$ as the grid:

$$\Omega = \left\{ \left( \frac{i_1}{p}, \ldots, \frac{i_{N_0}}{p} \right) \mid 0 \leq i_j \leq p, i_j \in \mathbb{N}, \forall 1 \leq j \leq N_0 \right\},$$

565 so that the cardinality of $\Omega = \{x_i\}_{i=1}^{P}$ is $P := (p+1)^{N_0}$. Similar to the sketch proof, consider
566 $\mathbf{D} := \left[ f(x_1), f(x_2), \ldots, f(x_P) \right]$. Our goal is to prove that $\mathbf{D} \in \overline{\mathcal{F}_{\mathbf{I}}(\Omega)} \setminus \mathcal{F}_{\mathbf{I}}(\Omega)$.

567 First, notice that $\mathbf{D} \in \overline{\mathcal{F}_{\mathbf{I}}(\Omega)}$ as an immediate consequence of Lemma B.2 and Lemma B.1.

568 It remains to show that $\mathbf{D} \notin \mathcal{F}_{\mathbf{I}}(\Omega)$. We proceed by contradiction, assuming that there exists $\theta \in \mathcal{N}_{\mathbf{I}}$
569 such that $\mathcal{R}_\theta(\Omega) = \mathbf{D}$.

570 To show the contradiction, we start by showing that, as a consequence of Lemma B.4 there exists
571 $x \in \Omega$ such that the hypercube whose vertices are the $2^{N_0}$ points

$$\left\{ x + \left( \frac{i_1}{p}, \ldots, \frac{i_{N_0}}{p} \right) \mid i_j \in \{0,1\}, \forall 1 \leq j \leq N_0 \right\} \subseteq \Omega, \tag{12}$$

572 lies entirely inside a linear region $\mathcal{P}$ of the continuous piecewise linear function $\mathcal{R}_\theta$ [1]. Denote
573 $K = 2^{\sum_{i=1}^{L} N_i}$ a bound on the number of such linear regions, see e.g. [24]. Each frontier between a
574 pair of linear regions can be completed into a hyperplane, leading to at most $H = K^2$ hyperplanes.
575 Since $p = 3N_0 K^2 \geq 3N_0 H$, by Lemma B.4 there exists $x \in \Omega$ such that the claimed hypercube lies
576 entirely inside a polytope delimited by these hyperplanes. As this polytope is itself included in some
577 linear region $\mathcal{P}$ of $\mathcal{R}_\theta$, this establishes our intermediate claim.

Now, define $v_0 := x$ and $v_i := x + (1/p)\mathbf{e}_i, i \in [\![N_0]\!]$ where $\mathbf{e}_i$ is the $i$th canonical vector. Denote $\mathbf{P} \in \mathbb{R}^{N_L \times N_0}$ the matrix such that the restriction of $\mathcal{R}_\theta$ to the piece $\mathcal{P}$ is $f_\mathcal{P}(x) = \mathbf{P}x + \mathbf{b}$. Since $\mathbf{P}$ is the Jacobian matrix of $\mathcal{R}_\theta$ in the linear region $\mathcal{P}$, we deduce from Lemma B.3 that $\mathbf{P} \in \mathcal{L}_{\mathbf{I}}$. Since the points $v_i$ belong to the hypercube which is both included in $\mathcal{P}$ and in $\Omega$ we have for each $i$:

$$\begin{aligned}
\mathbf{P}(v_0 - v_i) &= f_\mathcal{P}(v_0) - f_\mathcal{P}(v_i) \\
&= \mathcal{R}_\theta(v_0) - \mathcal{R}_\theta(v_i) \\
&= f(v_0) - f(v_i) \\
&= \mathbf{A}(v_0 - v_i).
\end{aligned}$$

578 where the third equality follows from the definition of $\mathbf{D}$ and the fact that we assume $\mathcal{R}_\theta(\Omega) = \mathbf{D}$.
579 Since $v_0 - v_i = \mathbf{e}_i/p, i = 1, \ldots, n$ are linearly independent, we conclude that $\mathbf{P} = \mathbf{A}$. This implies
580 $\mathbf{A} \in \mathcal{L}_{\mathbf{I}}$, hence the contradiction. This concludes the proof. $\qquad\square$

581 We now prove the intermediate technical lemmas.

*Proof of Lemma B.2.* Since $f \in \overline{\mathcal{F}_{\mathbf{I}}(\Omega')}$, there exists a sequence $\{\theta_k\}_{k \in \mathbb{N}}$ such that:

$$\lim_{k \to \infty} \sup_{x \in \Omega'} \| f(x) - \mathcal{R}_{\theta_k}(x) \| = 0$$

582 Denoting $\mathbf{D}_k := \left[ \mathcal{R}_{\theta_k}(x_1) \ldots \mathcal{R}_{\theta_k}(x_r) \right]$, since $x_i \in \Omega \subseteq \Omega', i = 1, \ldots, P$, it follows that $\mathbf{D}_k$
583 converges to $\mathbf{D}$. Since $\mathbf{D}_k \in \mathcal{F}_{\mathbf{I}}(\Omega)$ by construction, we get that $\mathbf{D} \in \overline{\mathcal{F}_{\mathbf{I}}(\Omega)}$. $\qquad\square$

*Proof of Lemma B.3.* For any $\theta \in \mathbf{I}$, $\mathcal{R}_\theta$ is a continuous piecewise linear function since it is the realization of a ReLU neural network [1]. Consider $\mathcal{P}$ a linear region of $\mathcal{R}_\theta$ with non-empty interior. The Jacobian matrix of $\mathcal{P}$ has the following form [28, Lemma 9]:

$$\mathbf{J} = \mathbf{W}_L \mathbf{D}_{L-1} \mathbf{W}_{L-1} \mathbf{D}_{L-2} \ldots \mathbf{D}_1 \mathbf{W}_1$$

584 where $\mathbf{D}_i$ is a binary diagonal matrix (diagonal matrix whose coefficients are either one or zero).
585 Since $\mathrm{supp}(\mathbf{D}_i \mathbf{W}_i) \subseteq \mathrm{supp}(\mathbf{W}_i) \subseteq I_i$, we have: $\mathbf{J} = \mathbf{W}_L \prod_{i=1}^{L-1} (\mathbf{D}_i \mathbf{W}_i) \in \mathcal{L}_I$. $\qquad\square$

*Proof of Lemma B.4.* Every edge of an elementary hypercube can be written as:

$$\left(x, x + \frac{1}{p}\mathbf{e}_i\right), x \in \Omega_p^N$$

where $\mathbf{e}_i$ is the $i$th canonical vector, $1 \leq i \leq N$. The points $x$ and $x + (1/p)\mathbf{e}_i$ are two *endpoints*. Note that in this proof we use the notation $(a, b)$ to denote the line segment whose endpoints are $a$ and $b$. By construction, $\Omega_p^N$ contains $p^N$ such elementary hypercubes. Given a collection of $H$ hyperplanes, we say that an elementary hypercube is an *intersecting hypercube* if it does not lie entirely inside a polytope generated by the hyperplanes, meaning that there exists a hyperplane that *intersects* at least one of its edges. More specifically, an edge and a hyperplane intersect if they have *exactly* one common point. We exclude the case where there are more than two common points since that implies that the edge lies completely in the hyperplane. The edges that are intersected by at least one hyperplane are called *intersecting edges*. Note that a hypercube can have intersecting edges, but it may not be an intersecting one. A visual illustration of this idea is presented in Figure 3.

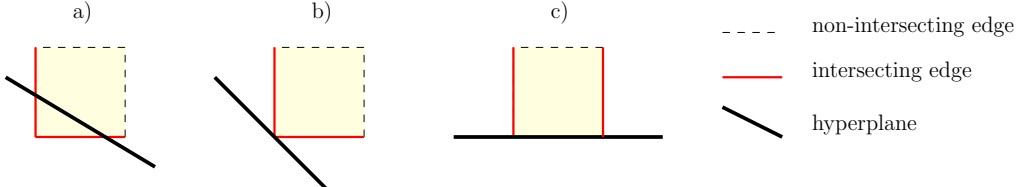

Figure 3: Illustration of definitions in $\mathbb{R}^2$: a) an intersecting hypercube with two intersecting edges; b) *not* an intersecting hypercube, but it has two intersecting edges; c) *not* an intersecting hypercube and it only has two intersecting edges (not three according to our definitions: the bottom edge is *not* intersecting).

Formally, a hyperplane $\{w^\top x + b = 0\}$ for $w \in \mathbb{R}^N$ and $b \in \mathbb{R}$ intersects an edge $(x, x + \frac{1}{p}\mathbf{e}_i)$ if:

$$\begin{cases} (w^\top x + b)\left[w^\top(x + \frac{1}{p}\mathbf{e}_i) + b\right] \leq 0 \\ \text{and} \\ w^\top x + b \neq 0 \text{ or } w^\top(x + \frac{1}{p}\mathbf{e}_i) + b \neq 0 \end{cases} \tag{13}$$

We further illustrate these notions in Figure 4. We emphasize that according to Equation (13), $\ell_3$ in Figure 4 does not intersect any edge *along its direction*.

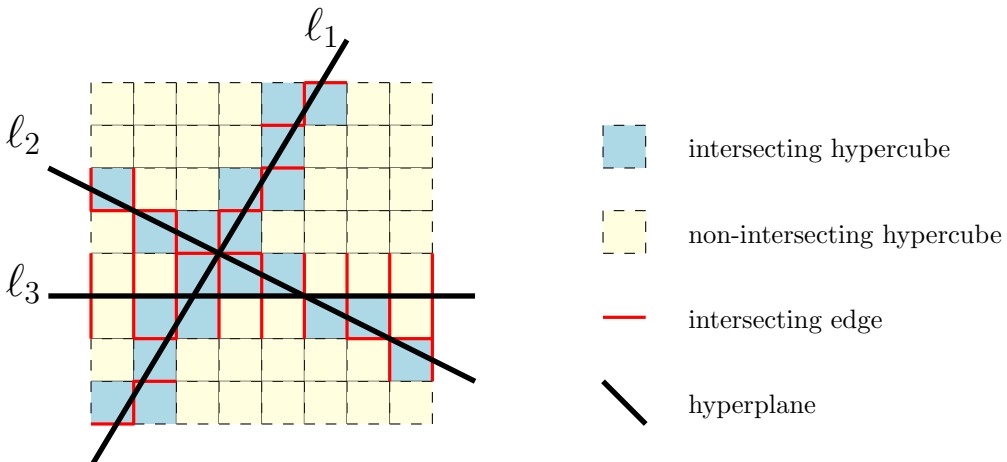

Figure 4: Illustration of intersecting hypercubes and hyperplanes in $\mathbb{R}^2$.

Clearly, the number of intersecting hypercubes is upper bounded by the number of intersecting edges. The rest of the proof is devoted to showing that this number is strictly smaller than $p^N$ *if $p \geq 3NH$*, as this will imply the existence of at least one non-intersecting hypercube.

To estimate the *maximum* number of intersecting edges, we analyze the *maximum* number of edges that a given hyperplane can intersect. For a fixed index $1 \leq i \leq N$, we count the number of edges of the form $(x, x + \frac{1}{p}\mathbf{e}_i)$ intersected by a single hyperplane. The key observation is: if we fix all the coordinates of $x$ except the $i$th one, then the edges $(x, x + \frac{1}{p}\mathbf{e}_i)$ form a line in the ambient space. Among those edges, there are at most *two* intersecting edges with respect to the given hyperplane. This happens only when the hyperplane intersects an edge at one of its endpoints (e.g., the hyperplane $\ell_2$ and the second vertical line in Figure 4). In total, for each $1 \leq i \leq N$ and each given hyperplane, there are at most $2(p+1)^{N-1}$ intersecting edges of the form $(x, x + \frac{1}{p}\mathbf{e}_i)$. For a given hyperplane, there are thus at most $2N(p+1)^{N-1}$ intersecting edges in total (since $i \in [\![N]\!]$). Since the number of hyperplanes is at most $H$, there are at most $2NH(p+1)^{N-1}$ intersecting edges, and this quantity also bounds the number of intersecting cubes as we have seen. With the assumption $p \geq 3NH$, we conclude by proving that $p^N > 2NH(p+1)^{N-1}$. Indeed, we have:

$$\frac{2NH(p+1)^{N-1}}{p^N} = \frac{2NH}{p}\left(\frac{p+1}{p}\right)^{N-1} = \frac{2NH}{p}\left(1 + \frac{1}{p}\right)^{N-1} < \frac{2NH}{p}\left(1 + \frac{1}{p}\right)^{NH}$$

$$\leq \frac{2NH}{3NH}\left(1 + \frac{1}{3NH}\right)^{NH} \leq \frac{2e^{1/3}}{3} \approx 0.93 < 1$$

where we used that $(1 + 1/n)^n \leq e \approx 2.71828$, the Euler number. $\square$

## B.3  Proof of Theorem 3.4

*Proof.* We denote $\mathbf{X} = [x_1, \ldots, x_P] \in \mathbb{R}^{N_0 \times P}$, the matrix representation of $\Omega$. Our proof has three main steps:

**Step 1:**  We show that we can reduce the study of the closedness of $\mathcal{F}_{\mathbf{I}}(\Omega)$ to that of the closedness of a union of subsets of $\mathbb{R}^P$, associated to the vectors $\mathbf{W}_2$. To do this, we prove that for any element $f \in \mathcal{F}_{\mathbf{I}}(\Omega)$, there exists a set of parameters $\theta \in \mathcal{N}_{\mathbf{I}}$ such that the matrix of the second layer $\mathbf{W}_2$ belongs to $\{-1, 0, 1\}^{1 \times N_1}$ (since we assume $N_2 = 1$). This idea is reused from the proof of [1, Theorem 4.1].

For $\theta := \{(\mathbf{W}_i, \mathbf{b}_i)_{i=1}^2\} \in \mathcal{N}_{\mathbf{I}}$, the function $\mathcal{R}(\theta)$ has the form:

$$\mathcal{R}_\theta(x) = \mathbf{W}_2\sigma(\mathbf{W}_1 x + \mathbf{b}_1) + \mathbf{b}_2 = \sum_{i=1}^{N_1} \mathbf{w}_{2,i}\sigma(\mathbf{w}_{1,i} x + \mathbf{b}_{1,i}) + \mathbf{b}_2$$

where $\mathbf{w}_{1,i} = \mathbf{W}_1[i, :] \in \mathbb{R}^{1 \times N_0}, \mathbf{w}_{2,i} = \mathbf{W}_2[i] \in \mathbb{R}, \mathbf{b}_{1,i} = \mathbf{b}[i] \in \mathbb{R}$. Moreover, if $w_{2,i}$ is different from zero, we have:

$$\mathbf{w}_{2,i}\sigma(\mathbf{w}_{1,i} x + \mathbf{b}_1) = \frac{\mathbf{w}_{2,i}}{|\mathbf{w}_{2,i}|}\sigma(|\mathbf{w}_{2,i}|\mathbf{w}_{1,i} x + |\mathbf{w}_{2,i}|\mathbf{b}_{1,i}).$$

In that case, one can assume that $\mathbf{w}_{2,i}$ can be equal to either $-1$ or $1$. Thus, we can assume $\mathbf{w}_{2,i} \in \{\pm 1, 0\}$. For a vector $\mathbf{v} \in \{-1, 0, 1\}^{1 \times N_1}$, we define:

$$F_{\mathbf{v}} = \{[\mathcal{R}_\theta(x_1), \ldots, \mathcal{R}_\theta(x_P)] \mid \theta \in \mathcal{N}_{\mathbf{I}, \mathbf{v}}\} \tag{14}$$

where $\mathcal{N}_{\mathbf{I}, \mathbf{v}} \subseteq \mathcal{N}_{\mathbf{I}}$ is the set of $\theta = \{(\mathbf{W}_i, \mathbf{b}_i)_{i=1}^2\}$ with $\mathbf{W}_2 = \mathbf{v} \in \{0, 1\}^{1 \times N_1}$, i.e., in words, $F_{\mathbf{v}}$ is the image of $\Omega$ through the function $\mathcal{R}_\theta, \theta \in \mathcal{N}_{\mathbf{I}, \mathbf{v}}$.

Define $\mathbb{V} := \{\mathbf{v} \mid \mathrm{supp}(\mathbf{v}) \subseteq I_2\} \cap \{0, \pm 1\}^{1 \times N_1}$. Clearly, for $\mathbf{v} \in \mathbb{V}$, $F_{\mathbf{v}} \subseteq \mathcal{F}_{\mathbf{I}}(\Omega)$. Therefore,

$$\bigcup_{\mathbf{v} \in \mathbb{V}} F_{\mathbf{v}} \subseteq \mathcal{F}_{\mathbf{I}}(\Omega).$$

Moreover, by our previous argument, we also have:

$$\mathcal{F}_{\mathbf{I}}(\Omega) \subseteq \bigcup_{\mathbf{v} \in \mathbb{V}} F_{\mathbf{v}}.$$

Therefore,

$$\mathcal{F}_{\mathbf{I}}(\Omega) = \bigcup_{\mathbf{v} \in \mathbb{V}} F_{\mathbf{v}}.$$

**Step 2:** Using the first step, to prove that $\mathcal{F}_{\mathbf{I}}(\Omega)$ is closed, it is sufficient to prove that $F_{\mathbf{v}}$ is closed, $\forall \mathbf{v} \in \mathbb{V}$. This can be accomplished by further decomposing $F_{\mathbf{v}}$ into smaller closed sets. We denote $\theta'$ the set of parameters $\mathbf{W}_1, \mathbf{b}_1$ and $\mathbf{b}_2$. In the following, only the parameters of $\theta'$ are varied since $\mathbf{W}_2$ is now fixed to $\mathbf{v}$.

Due to the activation function $\sigma$, for a given data point $x_j \in \Omega$, we have:

$$\sigma(\mathbf{W}x_j + \mathbf{b}_1) = \mathbf{D}_j(\mathbf{W}x_j + \mathbf{b}_1) \tag{15}$$

where $\mathbf{D}_j \in \mathcal{D}$, the set of binary diagonal matrices, and its diagonal coefficients $\mathbf{D}_j[i,i]$ are determined by:

$$\mathbf{D}_j[i,i] = \begin{cases} 0 & \text{if } \mathbf{W}[i,:]x_j + \mathbf{b}_1[i] \leq 0 \\ 1 & \text{if } \mathbf{W}[i,:]x_j + \mathbf{b}_1[i] \geq 0 \end{cases}. \tag{16}$$

Note that $\mathbf{D}_j[i,i]$ can take both values $0$ or $1$ if $\mathbf{W}[i,:]x_j + \mathbf{b}_1[i] = 0$. We call the matrix $\mathbf{D}_j$ the activation matrix of $x_j$. Therefore, for (15) to hold, the $N_1$ constraints of the form (16) must hold simultaneously. It is important to notice that all these constraints are linear w.r.t. $\theta'$. We denote $\mathbf{z}$ a vectorized version of $\theta'$ (i.e., we concatenate all coefficients whose indices are in $I_1$ of $\mathbf{W}$ and $\mathbf{b}_1, \mathbf{b}_2$ into a long vector), and we write all the constraints in (15) in a compact form:

$$\mathcal{A}(\mathbf{D}_j, x_j)\mathbf{z} \leq \mathbf{0}_{N_1}$$

where $\mathcal{A}(\mathbf{D}_j, x_j)$ is a constant matrix that depend on $\mathbf{D}_j$ and $x_j$.

Set $\theta = (\mathbf{v}, \mathbf{z})$. Given that (15) holds, we deduce that:

$$\mathcal{R}_\theta(x_j) = \mathbf{v}\sigma(\mathbf{W}x_j + \mathbf{b}_1) + \mathbf{b}_2 = \mathbf{v}\mathbf{D}_j(\mathbf{W}x_j + \mathbf{b}_1) + \mathbf{b}_2 = \mathcal{V}(\mathbf{D}_j, x_j, \mathbf{v})\mathbf{z}$$

where $\mathcal{V}(\mathbf{D}_j, x_j, \mathbf{v})$ is a constant matrix that depends on $\mathbf{D}_j, \mathbf{v}, x_j$. In particular, $\mathcal{R}_\theta(x_j)$ is also a linear function w.r.t the parameters $\mathbf{z}$ . Assume that the activation matrices of $(x_1, \ldots, x_P)$ are $(\mathbf{D}_1, \ldots, \mathbf{D}_P)$, then we have:

$$\mathcal{R}_\theta(\Omega) = (\mathcal{V}(\mathbf{D}_1, x_1, \mathbf{v})\mathbf{z}, \ldots, \mathcal{V}(\mathbf{D}_P, x_P, \mathbf{v})\mathbf{z}) \in \mathbb{R}^{1 \times P}.$$

To emphasize that $\mathcal{R}_\theta(\Omega)$ depends linearly on $\mathbf{z}$, for the rest of the proof, we will write $\mathcal{R}_\theta(\Omega)$ as a vector of size $P$ (instead of a row matrix $1 \times P$) as follows:

$$\mathcal{R}_\theta(\Omega) = \mathcal{V}(\mathbf{D}_1, \ldots, \mathbf{D}_P)\mathbf{z} \quad \text{where} \quad \mathcal{V}(\mathbf{D}_1, \ldots, \mathbf{D}_P) = \begin{pmatrix} \mathcal{V}(\mathbf{D}_1, x_1, \mathbf{v}) \\ \vdots \\ \mathcal{V}(\mathbf{D}_P, x_P, \mathbf{v}) \end{pmatrix}.$$

Moreover, to have $(\mathbf{D}_1, \ldots, \mathbf{D}_P)$ activation matrices, the parameters $\mathbf{z}$ need to satisfy:

$$\mathcal{A}(\mathbf{D}_1, \ldots, \mathbf{D}_P)\mathbf{z} \leq \mathbf{0}_Q$$

where $Q = PN_1$ and

$$\mathcal{A}(\mathbf{D}_1, \ldots, \mathbf{D}_P) = \begin{pmatrix} \mathcal{A}(\mathbf{D}_1, x_1) \\ \vdots \\ \mathcal{A}(\mathbf{D}_P, x_P) \end{pmatrix}.$$

Thus, the set of $\mathcal{R}_\theta(\Omega)$ *given* the activation matrices $(\mathbf{D}_1, \ldots, \mathbf{D}_P)$ has the following compact form:

$$F_{\mathbf{v}}^{(\mathbf{D}_1, \ldots, \mathbf{D}_P)} := \{\mathcal{V}(\mathbf{D}_1, \ldots, \mathbf{D}_P)\mathbf{z} \mid \mathcal{A}(\mathbf{D}_1, \ldots, \mathbf{D}_P)\mathbf{z} \leq \mathbf{0}\}.$$

Clearly, $F_{\mathbf{v}}^{(\mathbf{D}_1, \ldots, \mathbf{D}_P)} \subseteq F_{\mathbf{v}}$ since each element is equal to $\mathcal{R}_\theta(\Omega)$ with $\theta = (\mathbf{v}, \mathbf{z})$ for some $\mathbf{z}$. On the other hand, each element of $F_{\mathbf{v}}$ is an element of $F_{\mathbf{v}}^{(\mathbf{D}_1, \ldots, \mathbf{D}_P)}$ for some $(\mathbf{D}_1, \ldots, \mathbf{D}_P) \in \mathcal{D}^P$ since the set of activation matrices corresponding to any $\theta$ is in $\mathcal{D}^P$. Therefore,

$$F_{\mathbf{v}} = \bigcup_{(\mathbf{D}_1, \ldots, \mathbf{D}_P) \in \mathcal{D}^P} F_{\mathbf{v}}^{(\mathbf{D}_1, \ldots, \mathbf{D}_P)}.$$

**Step 3:** Using the previous step, it is sufficient to prove that $F_{\mathbf{v}}^{(\mathbf{D}_1,\ldots,\mathbf{D}_P)}$ is closed, for any $\mathbf{v}, (\mathbf{D}_1,\ldots,\mathbf{D}_P) \in \mathcal{D}^P$. To do so, we write $F_{\mathbf{v}}^{(\mathbf{D}_1,\ldots,\mathbf{D}_P)}$ in a more general form:

$$\{\mathbf{Az} \mid \mathbf{Cz} \leq \mathbf{y}\}. \tag{17}$$

Therefore, it is sufficient to prove that a set as in Equation (17) is closed. These sets are linear transformations of an intersection of a finite number of half-spaces. Since the intersection of a finite number of halfspaces is *stable* under linear transformations (cf. Lemma B.5 below), and the intersection of a finite number of half-spaces is a closed set itself, the proof can be concluded. $\quad\square$

**Lemma B.5** (Closure of intersection of half-spaces under linear transformations). *For any* $\mathbf{A} \in \mathbb{R}^{m \times n}, \mathbf{C} \in \mathbb{R}^{\ell \times n}, \mathbf{y} \in \mathbb{R}^{\ell}$, *there exists* $\mathbf{C}' \in \mathbf{R}^{k \times m}, \mathbf{b}' \in \mathbf{R}^k$ *such that:*

$$\{\mathbf{Ax} \mid \mathbf{Cx} \leq \mathbf{y}\} = \{\mathbf{C}'\mathbf{z} \leq \mathbf{b}'\}.$$

*Proof.* The proof uses Fourier–Motzkin elimination [4]. This method is a quantifier elimination algorithm for linear functions [5]. In fact, the LHS can be written as: $\{\mathbf{t} \mid \mathbf{t} = \mathbf{Ax}, \mathbf{Cx} \leq \mathbf{y}\}$, or more generally,

$$\left\{ \mathbf{t} \mid \exists \mathbf{x} \in \mathbb{R}^n \text{ s.t. } \mathbf{B}\begin{pmatrix}\mathbf{x} \\ \mathbf{t}\end{pmatrix} \leq \mathbf{v} \right\} \subseteq \mathbb{R}^m$$

where $\begin{pmatrix}\mathbf{x} \\ \mathbf{t}\end{pmatrix}$ is the concatenation of two vectors $(\mathbf{x}, \mathbf{t})$ and the linear constraints imposed by $\mathbf{B}\begin{pmatrix}\mathbf{x} \\ \mathbf{t}\end{pmatrix} \leq \mathbf{v}$ replace the two linear constraints $\mathbf{Cx} \leq \mathbf{y}$ and $\mathbf{t} = \mathbf{Ax}$. The idea is to show that:

$$\left\{ \mathbf{t} \mid \exists \mathbf{x} \in \mathbb{R}^n \text{ s.t. } \mathbf{B}\begin{pmatrix}\mathbf{x} \\ \mathbf{t}\end{pmatrix} \leq \mathbf{v} \right\} = \left\{ \mathbf{t} \mid \exists \mathbf{x}' \in \mathbb{R}^{n-1} \text{ s.t. } \mathbf{B}'\begin{pmatrix}\mathbf{x}' \\ \mathbf{t}\end{pmatrix} \leq \mathbf{v}' \right\} \tag{18}$$

for some matrix $\mathbf{B}'$ and vector $\mathbf{v}'$. By doing so, we reduce the dimension of the quantified parameter $\mathbf{x}$ by one. By repeating this procedure until there is no more quantifier, we prove the lemma. The rest of this proof is thus devoted to show that $\mathbf{B}', \mathbf{v}'$ as in (18) do exist.

We will show how to eliminate the first coordinate of $\mathbf{x}[1]$. First, we partition the set of linear constraints of LHS of (18) into three groups:

1. $S_0 := \{j \mid \mathbf{B}[j,1] = 0\}$: In this case, $\mathbf{x}[1]$ does not appear in this constraint, there is nothing to do.

2. $S_+ := \{j \mid \mathbf{B}[j,1] > 0\}$, for $j \in S_+$, we can rewrite the constraints $\mathbf{B}[j,:]\begin{pmatrix}\mathbf{x} \\ \mathbf{t}\end{pmatrix} \leq \mathbf{v}[j]$ as:

$$\mathbf{x}[1] \leq \gamma[j] + \sum_{i=2}^n \alpha[i]\mathbf{x}[i] + \sum_{i=1}^m \beta[i]\mathbf{t}[i] := B_j^+(\mathbf{x}', \mathbf{t})$$

for some suitable $\gamma[j], \alpha[i], \beta[i]$ where $\mathbf{x}'$ is the last $(n-1)$ coordinate of the vector $\mathbf{x}$.

3. $S_- := \{j \mid \mathbf{B}[j,1] < 0\}$: for $j \in S_-$, we can rewrite the constraints $\mathbf{B}[j,:]\begin{pmatrix}\mathbf{x} \\ \mathbf{t}\end{pmatrix} \leq \mathbf{v}_j$ as:

$$\mathbf{x}[1] \geq \gamma[j] + \sum_{i=2}^n \alpha[i]\mathbf{x}[i] + \sum_{i=1}^m \beta[i]\mathbf{t}[i] := B_j^-(\mathbf{x}', \mathbf{t}).$$

For the existence of such $\mathbf{x}[1]$, it is necessary and sufficient that:

$$B_k^+(\mathbf{x}', \mathbf{t}) \geq B_j^-(\mathbf{x}', \mathbf{t}), \qquad \forall k \in S_+, j \in S_-. \tag{19}$$

Thus, we form the matrix $\mathbf{B}'$ and the vector $\mathbf{v}'$ such that the linear constraints written in the following form:

$$\mathbf{B}'\begin{pmatrix}\mathbf{x}' \\ \mathbf{t}\end{pmatrix} \leq \mathbf{v}'$$

represent all the linear constraints in the set $S_0$ and those in the form of (19). Using this procedure recursively, one can eliminate all quantifiers and prove the lemma. $\quad\square$

---

[4]More detail about this method can be found in this link

[5]In fact, the algorithm determining the closedness of $\mathcal{L}_{\mathbf{I}}$ is also a quantifier elimination one, but it can be used in a more general setting: polynomials

## B.4 Proofs for Lemma 3.3

Since we use tools of real algebraic geometry, this section provides basic notions of real algebraic geometry for readers who are not familiar with this domain. It is organized and presented as in the textbook [2] (with slight modifications to better suit our needs). For a more complete presentation, we refer readers to [2, Chapter 2].

**Definition B.1** (Semi-algebraic sets). *A semi-algebraic set of $\mathbb{R}^n$ has the form:*

$$\bigcup_{i=1}^{k}\{x \in \mathbb{R}^n \mid P_i(x) = 0 \wedge \bigwedge_{j=1}^{\ell_i} Q_{i,j}(x) > 0\}$$

*where $P_i, Q_{i,j} : \mathbb{R}^n \mapsto \mathbb{R}$ are polynomials and $\wedge$ is the "and" logic.*

The following theorem is known as the projection theorem of semi-algebraic sets. In words, the theorem states that: The projection of a semi-algebraic set to a lower dimension is still a semi-algebraic set (of lower dimension).

**Theorem B.6** (Projection theorem of semi-algebraic sets [2, Theorem 2.92]). *Let $A$ be a semi-algebraic set of $\mathbb{R}^n$ and define:*

$$B = \{(x_1, \ldots, x_{n-1}) \mid \exists x_n, (x_1, \ldots, x_{n-1}, x_n) \in A\}$$

*then $B$ is a semi-algebraic set of $\mathbb{R}^{n-1}$.*

Theorem B.6 is a powerful result. Its proof [2, Section 2.4] (which is constructive) shows a way to express $B$ (in Theorem B.6) by using only the first $n-1$ variables $(x_1, \ldots, x_{n-1})$.

Next, we introduce the language of an ordered field and sentence. Readers which are not familiar to the notion of ordered field can simply think of it as $\mathbb{R}$ and its subring as $\mathbb{Q}$. Example for fields that is not ordered is $\mathbb{C}$ (we cannot compare two arbitrary complex number). Therefore, the notion of semi-algebraic set in Definition B.1 (which contains $Q_{i,j}(x) > 0$) does not make sense when the underlying field is not ordered.

The central definition of the language of $\mathbb{R}$ is *formula*, an abstraction of semi-algebraic sets. In particular, the definition of formula is recursive: formula is built from atoms - equalities and inequalities of polynomials whose coefficients are in a subring $\mathbb{Q}$ of $\mathbb{R}$. It can be also formed by combining with logical connectives "and", "or", and "negation" ($\wedge, \vee, \neg$) and existential/universal quantifiers ($\exists, \forall$). Formula has variables, which are those of atoms in the formula itself. *Free variables* of a formula are those which are not preceded by a quantifier ($\exists, \forall$). The definitions of a formula and its free variables are given recursively as follow:

**Definition B.2** (Language of the ordered field with coefficients in a ring). *Consider $\mathbf{R}$ an ordered field and $\mathbf{Q} \subseteq \mathbf{R}$ a subring, a formula $\Phi$ and its set of free variables $\mathtt{Free}(X)$ are defined recursively as:*

1. *An atom: if $P \in \mathbf{Q}[X]$ (where $\mathbf{Q}[X]$ is the set of polynomials with coefficients in $\mathbf{Q}$) then $\Phi := (P = 0)$ (resp. $\Phi := (P > 0)$) is a formula and its set of free variables is $\mathtt{Free}(\Phi) := \{X_1, \ldots, X_n\}$ where $n$ is the number of variables.*

2. *If $\Phi_1$ and $\Phi_2$ are formulas, then so are $\Phi_1 \vee \Phi_2, \Phi_1 \wedge \Phi_2$ and $\neg\Phi_1$. The set of free variables is defined as:*

   *(a) $\mathtt{Free}(\Phi_1 \vee \Phi_2) := \mathtt{Free}(\Phi_1) \cup \mathtt{Free}(\Phi_2)$.*
   *(b) $\mathtt{Free}(\Phi_1 \wedge \Phi_2) := \mathtt{Free}(\Phi_1) \cup \mathtt{Free}(\Phi_2)$.*
   *(c) $\mathtt{Free}(\neg\Phi_1) = \mathtt{Free}(\Phi_1)$.*

3. *If $\Phi$ is a formula and $X \in \mathtt{Free}(\Phi)$, then $\Phi' = (\exists X)\Phi$ and $\Phi'' = (\forall X)\Phi$ are also formulas and $\mathtt{Free}(\Phi') := \mathtt{Free}(\Phi) \setminus \{X\}$, and $\mathtt{Free}(\Phi'') := \mathtt{Free}(\Phi) \setminus \{X\}$.*

**Definition B.3** (Sentence). *A sentence is a formula of an ordered field with no free variable.*

**Example B.1.** *Consider two formulas:*

$$\Phi_1 = \{\exists X_1, X_1^2 + X_2^2 = 0\}$$
$$\Phi_2 = \{\exists X_1, \exists X_2, X_1^2 + X_2^2 = 0\}$$

While $\Phi_1$ is a normal formula, $\Phi_2$ is a sentence and given an underlying field ($\mathbb{R}$, for instance), $\Phi_2$ is either correct or not. Here, $\Phi_2$ is correct (since $X_1^2 + X_2^2 = 0$ has a root $(0,0)$). Nevertheless, if one consider $\Phi_2' = \{\exists X_1, \exists X_2, X_1^2 + X_2^2 = -1\}$, then $\Phi_2'$ is not correct.

An algorithm deciding whether a sentence is correct or not is very tempting since formula and sentence can be used to express many theorems in the language of an ordered field. The proof or disproof will be then given by an algorithm. Such an algorithm does exist, as follow:

**Theorem B.7** (Decision problem [2, Algorithm 11.36]). *There exists an algorithm to decide whether a given sentence is correct is not with complexity $O(sd)^{O(1)^{k-1}}$ where $s$ is the bound on the number of polynomials in $\Phi$, $d$ is the bound on the degrees of the polynomials in $\Phi$ and $k$ is the number of variables.*

A full description of [2, Algorithm 11.36] (quantifier elimination algorithm) is totally out of the scope of this paper. Nevertheless, we will try to explain it in a concise way. The key observation is Theorem B.6, the central result of real algebraic geometry. As discussed right after Theorem B.6, its proof implies that one can replace a sentence by another whose number of quantifiers is reduced by one such that both sentences agree (both are true or false). Applying this procedure iteratively will result into a sentence without any variable (and the remain are only *coefficients in the subring*). We check the correctness of this final sentence by trivially verifying all the equalities/inequalities and obtain the answer for the original one.

*Proof of Lemma 3.3.* To decide whether $\mathcal{L}_\mathbf{I}$ is closed or not, it is equivalent to decide if the following *sentence* (see Definition B.3) is true or false:

$$\exists \mathbf{A}, (\forall \mathbf{X}_L, \dots, \mathbf{X}_1, P(\mathbf{A}, \mathbf{X}_L, \dots, \mathbf{X}_1) > 0) \wedge$$
$$(\forall \epsilon > 0, \exists \mathbf{X}_L', \dots, \mathbf{X}_1', P(\mathbf{A}, \mathbf{X}_L', \dots, \mathbf{X}_1') - \epsilon < 0)$$

where $P(\mathbf{A}, \mathbf{X}_1, \dots, \mathbf{X}_L) := \sum_{(i,j)} (\mathbf{A}[i,j] - P_{i,j}^\mathbf{I}(\mathbf{X}_L, \dots, \mathbf{X}_1))^2$.

This sentence basically asks whether there exists a matrix $\mathbf{A} \in \overline{\mathcal{F}_\mathbf{I}} \setminus \mathcal{F}_\mathbf{I}$ or not. It can be proved that this sentence can be decided to be true or false using real algebraic geometry tools (see Theorem B.7), with a complexity $O\left((sd)^{C^{k-1}}\right)$ where $C$ is a universal constant and $s, d, k$ are the number of polynomials, the maximum degree of the polynomials and the number of variables in the sentence, respectively. Applying this to our case, we have $s = 2, d = 2L, k = N_L N_0 + 1 + 2\sum_{\ell=1}^L |I_\ell|$ (remind that $|I_\ell|$ is the total number of unmasked coefficients of $\mathbf{X}_\ell$). $\qquad\square$

### B.5 Polynomial algorithm to detect support constraints $\mathbf{I} = (I, J)$ with non-closed $\mathcal{L}_\mathbf{I}$.

The following sufficient condition for non-closedness is based on the existence in the support constraint of $2 \times 2$ blocks sharing the essential properties of a $2 \times 2$ LU support constraint.

**Lemma B.8.** *Consider a pair $\mathbf{I} = (I, J) \in \{0,1\}^{m \times r} \times \{0,1\}^{r \times n}$ of support constraints for the weight matrices of one-hidden-layer neural network. If there exists four indices $1 \le i_1, i_2 \le m, 1 \le j_1, j_2 \le n$ and two indices $k \ne l, 1 \le k, l \le r$ such that:*

*1. For each pair $(i,j) \in \{(i_1, j_1), (i_1, j_2), (i_2, j_1)\}$ we have:*

$$(i,j) \in \mathrm{supp}(I[:,k]J[k,:]) \text{ and } (i,j) \notin \mathrm{supp}(I[:,\ell]J[\ell,:]), \forall \ell \ne k.$$

*2. The pair $(i_2, j_2)$ belongs to $\mathrm{supp}(I[:,k]J[k,:])$ and to $\mathrm{supp}(I[:,l]J[l,:])$.*

*then $\mathcal{L}_\mathbf{I}$ is non-closed.*

*Proof.* First, it is easy to see that the assumptions of this lemma are equivalent to those of [18, Theorem 4.20] since $\mathrm{supp}(I[:,k]J[k,:])$ is precisely the $k$th rank-one support of the pair $(I, J)$ [18, Definition 3.1]. Without loss of generality, one can assume that $i_1, j_1 = 1, i_2, j_2 = 2$ and $k = 1, l = 2$. We will prove that $\mathbf{A} \in \overline{\mathcal{L}_\mathbf{I}} \setminus \mathcal{L}_\mathbf{I}$ where

$$\mathbf{A} := \begin{pmatrix} \mathbf{A}' & \mathbf{0} \\ \mathbf{0} & \mathbf{0} \end{pmatrix} \in \mathbb{R}^{m \times n}, \text{ with } \mathbf{A}' := \begin{pmatrix} 0 & 1 \\ 1 & 0 \end{pmatrix} \in \mathbb{R}^{2 \times 2}.$$

This can be shown in two steps:

1. Proof that $\mathbf{A} \in \overline{\mathcal{L}_{\mathbf{I}}}$: For any $\epsilon > 0$, consider two factors:

$$\mathbf{X}_\epsilon = \begin{pmatrix} \mathbf{X}'_\epsilon & \mathbf{0} \\ \mathbf{0} & \mathbf{0} \end{pmatrix}, \mathbf{Y}_\epsilon = \begin{pmatrix} \mathbf{Y}'_\epsilon & \mathbf{0} \\ \mathbf{0} & \mathbf{0} \end{pmatrix}$$

where $\mathbf{X}'_\epsilon, \mathbf{Y}'_\epsilon \in \mathbb{R}^{2\times 2}$ respect the support constraints corresponding to the **LU** architecture. It is not hard to see that such a construction of $(\mathbf{X}_\epsilon, \mathbf{Y}_\epsilon)$ satisfies the support constraints $(I, J)$ (due to the assumption of the lemma and the value of indices). Moreover, we also have:

$$\|\mathbf{A} - \mathbf{X}_\epsilon \mathbf{Y}_\epsilon\|_F = \|\mathbf{A}' - \mathbf{X}'_\epsilon \mathbf{Y}'_\epsilon\|_F$$

Thus, to have $\|\mathbf{A} - \mathbf{X}_\epsilon \mathbf{Y}_\epsilon\|_F \leq \epsilon$, it is sufficient to choose a pair of factors $(\mathbf{X}'_\epsilon, \mathbf{Y}'_\epsilon)$ respecting the **LU** architecture of size $2 \times 2$ such that $\|\mathbf{A}' - \mathbf{X}'_\epsilon \mathbf{Y}'_\epsilon\|_F \leq \epsilon$. Such a pair exists, since the set of matrices admitting the exact **LU** decomposition is dense in $\mathbb{R}^{2\times 2}$. This holds for any $\epsilon > 0$. Therefore, $\mathbf{A} \in \overline{\mathcal{L}_{\mathbf{I}}}$.

2. Proof that $\mathbf{A} \notin \mathcal{L}_{\mathbf{I}}$: Assume there exist a pair of factors $(\mathbf{X}, \mathbf{Y})$ whose product $\mathbf{XY} = \mathbf{A}$ and supports are included in $(I, J)$. Due to the assumptions on the pairs $(i_1, j_1), (i_1, j_2), (i_2, j_1)$, we must have:

$$\begin{cases} \mathbf{X}[1,1]\mathbf{Y}[1,1] &= \mathbf{A}[1,1] = 0 \\ \mathbf{X}[2,1]\mathbf{Y}[1,1] &= \mathbf{A}[2,1] = 1 \\ \mathbf{X}[1,1]\mathbf{Y}[1,2] &= \mathbf{A}[1,2] = 1. \end{cases}$$

It is easy to see that it is impossible. Therefore, $\mathbf{A} \notin \mathcal{L}_{\mathbf{I}}$. $\qquad\square$

Given a pair of support constraints $\mathbf{I}$, it is possible to check in time polynomial in $m, r, n$ whether the conditions of Lemma B.8 hold. A brute force algorithm has complexity $O(m^2 n^2 r)$. A more clever implementation with careful book-marking can reduce this complexity to $O(\min(m, n)mnr)$.

## C  Proofs for results in Section 4

### C.1  Proof of Theorem 4.1

In fact, Theorem 4.1 is a corollary of Lemma B.1. Thus, we will give a proof for Lemma B.1 in the following.

*Proof of Lemma B.1.* Since $\mathbf{A} \in \overline{\mathcal{L}_{\mathbf{I}}} \backslash \mathcal{L}_{\mathbf{I}} \subseteq \mathbb{R}^{N_L \times N_0}$, we have:

1. $\mathbf{A} \notin \mathcal{L}_{\mathbf{I}}$.

2. There exists a sequence $\{(\mathbf{X}_i^k)_{i=1}^L\}_{k\in\mathbb{N}}$ such that $\lim_{k\to\infty} \|\mathbf{X}_L^k \ldots \mathbf{X}_1^k - \mathbf{A}\| = 0$ for any norm defined on $\mathbb{R}^{N_0}$.

We will prove that the linear function: $f(x) := \mathbf{A}x$ satisfies $f \in \overline{\mathcal{F}_{\mathbf{I}}} \setminus \mathcal{F}_{\mathbf{I}}$ (where $\overline{\mathcal{F}_{\mathbf{I}}}$ is the closure of $\mathcal{F}_{\mathbf{I}}$ in $(C^0(\Omega), \|\cdot\|_\infty)$, that is to say $f$ is not the realization of any neural network but it is the uniform limit of the realizations of a sequence of neural networks). Firstly, we prove that $f \notin \mathcal{F}_{\mathbf{I}}$. For the sake of contradiction, assume there exists $\theta = (\mathbf{W}_i, \mathbf{b}_i)_{i=1}^L \in \mathcal{N}_{\mathbf{I}}$ such that $\mathcal{R}_\theta = f$. Since $\mathcal{R}_\theta$ is the realization of a ReLU neural network, it is a continuous piecewise linear function. Therefore, since $\Omega$ has non-empty interior, there exist a non-empty open subset $\Omega'$ of $\mathbb{R}^d$ such that $\Omega' \subseteq \Omega$ and $\mathcal{R}_\theta$ is linear on $\Omega'$, i.e., there are $\mathbf{A}' \in \mathbb{R}^{N_L \times N_0}, \mathbf{b} \in \mathbb{R}^{N_L}$ such that $\mathcal{R}_\theta(x) = \mathbf{A}'x + \mathbf{b}', \forall x \in \Omega'$. Since $f = \mathcal{R}_\theta$, we have: $\mathbf{A}' = \mathbf{A}$ and also equal to the Jacobian matrix of $\mathcal{R}_\theta$ on $\Omega'$. Using Lemma B.3 and the fact that $\mathbf{A} \notin \mathcal{L}_{\mathbf{I}}$, we conclude that $f \notin \mathcal{F}_{\mathbf{I}}$.

There remains to construct a sequence $\{\theta^k\}_{k\in\mathbb{N}}, \theta^k = (\mathbf{W}_i^k, \mathbf{b}_i^k)_{i=1}^L \in \mathcal{N}_{\mathbf{I}}$ such that $\lim_{k\to\infty} \|\mathcal{R}_{\theta^k} - f\|_\infty = 0$. We will rely on the sequence $\{(\mathbf{X}_i^k)_{i=1}^L\}_{k\in\mathbb{N}}$ for our construction. Given $k \in \mathbb{N}$ we simply define the weight matrices as $\mathbf{W}_i^k = \mathbf{X}_i^k, 1 \leq i \leq L$. The biases are built recursively. Starting from $c_1^k := \sup_{x\in\Omega} \|\mathbf{W}_1^k x\|_\infty$ and $\mathbf{b}_1^k := c_1^k \mathbf{1}_{N_1}$, we iteratively define for $2 \leq i \leq L-1$:

$$\gamma_{i-1}^k(x) := \mathbf{W}_{i-1}^k x + \mathbf{b}_{i-1}$$
$$c_i^k := \sup_{x\in\Omega} \|\gamma_{i-1}^k \circ \ldots \circ \gamma_1^k(x)\|_\infty$$
$$\mathbf{b}_i^k := c_i^k \mathbf{1}_{N_i}.$$

The boundedness of $\Omega$ ensures that $c_i^k$ is well-defined with a finite supremum. For $i = L$ we define:

$$\mathbf{b}_L^k = -\sum_{i=1}^{L-1} ( \prod_{j=i+1}^{L} \mathbf{W}_j)\mathbf{b}_i^k.$$

We will prove that $\mathcal{R}_{\theta^k}(x) = \left(\mathbf{X}_L^k \ldots \mathbf{X}_1^k\right) x, \forall x \in \Omega$. As a consequence, it is immediate that:

$$\lim_{k \to \infty} \|\mathcal{R}_{\theta^k} - f\|_\infty = \lim_{k \to \infty} \sup_{x \in \Omega} \|\mathcal{R}_{\theta^k}(x) - f(x)\|_2$$

$$\leq \lim_{k \to \infty} \|\mathbf{X}_L^k \ldots \mathbf{X}_1^k - \mathbf{A}\|_{2 \to 2} \sup_{x \in \Omega} \|x\|_2 = 0$$

where we used that all matrix norms are equivalent and denoted $\|\cdot\|_{2\to 2}$ the operator norm associated to Euclidean vector norms. Back to the proof that $\mathcal{R}_{\theta^k}(x) = \left(\mathbf{X}_L^k \ldots \mathbf{X}_1^k\right) x, \forall x \in \Omega$, due to our choice of $c_i^k$, we have for $2 \leq i \leq L - 1$:

$$\gamma_{i-1}^k \circ \ldots \circ \gamma_1^k(x) \geq 0, \forall x \in \Omega$$

where $\geq$ is taken in coordinate-wise manner. Therefore, an easy induction yields:

$$\begin{aligned}
\mathcal{R}_{\theta^k}(x) &= \gamma_L^k \circ \sigma \circ \gamma_{L-1}^k \circ \ldots \circ \sigma \circ \gamma_1^k(x) \\
&= \gamma_L^k \circ \gamma_{L-1}^k \ldots \circ \gamma_1^k(x) \\
&= \mathbf{W}_L^k(\ldots (\mathbf{W}_2^k(\mathbf{W}_1^k x + \mathbf{b}_1^k) + \mathbf{b}_2^k) \ldots) + \mathbf{b}_L^k \\
&= (\mathbf{X}_L^k \ldots \mathbf{X}_1^k)x + \sum_{i=1}^{L-1} ( \prod_{j=i+1}^{L} \mathbf{W}_j)\mathbf{b}_i^k - \sum_{i=1}^{L-1} ( \prod_{j=i+1}^{L} \mathbf{W}_j)\mathbf{b}_i^k \\
&= (\mathbf{X}_L^k \ldots \mathbf{X}_1^k)x.
\end{aligned}$$

$\square$

## C.2 Proof of Theorem 4.2

Given the involvement of Theorem 4.2, we decompose its proof and present it in two subsections: the first one establishes general results that do not use the assumption of Theorem 4.2. The second one combines the established results with the assumption of Theorem 4.2 to provide a full proof.

### C.2.1 Properties of the limit function of fixed support one-hidden-layer NNs

The main results of this parts are summarized in Lemma C.2 and Lemma C.3. It is important to emphasize that all results in this section do *not* make any assumption on $\mathbf{I}$.

We first introduce the following technical results.

**Lemma C.1** (Normalization of the rows of the first layer [26]). *Consider $\Omega$ a bounded subset of $\mathbb{R}^{N_0}$. Given any $\theta = \{(\mathbf{W}_i, \mathbf{b}_i)_{i=1}^2\} \in \mathcal{N}_{\mathbf{I}}$ and any norm $\|\cdot\|$ on $\mathbb{R}^{N_0}$, there exists $\tilde{\theta} := \{(\tilde{\mathbf{W}}_i, \tilde{\mathbf{b}}_i)_{i=1}^2\} \in \mathcal{N}_{\mathbf{I}}$ such that the matrix $\tilde{\mathbf{W}}_1$ has unit norm rows, $\|\tilde{\mathbf{b}}_1\|_\infty \leq C := \sup_{x \in \Omega} \sup_{\|u\| \leq 1} \langle u, x \rangle$ and $\mathcal{R}_\theta(x) = \mathcal{R}_{\tilde{\theta}}(x), \forall x \in \Omega$.*

*Proof.* We report this proof for self-completeness of this work. It is *not* a contribution, as it merely combines ideas from the proof of [26, Lemma D.2] and [26, Theorem 3.8, Steps 1-2].

We first show that for each set of weights $\theta \in \mathcal{N}_{\mathbf{I}}$ we can find another set of weights $\theta' = \{(\mathbf{W}_i', \mathbf{b}_i')_{i=1}^2\} \in \mathcal{N}_{\mathbf{I}}$ such that $\mathcal{R}_\theta = \mathcal{R}_{\theta'}$ on $\mathbb{R}^{N_0}$ and $\mathbf{W}_1'$ has unit norm rows. Note that $\|\mathbf{b}_1'\|_\infty$ can be larger than $C$. Indeed, given $\theta \in \mathcal{N}_{\mathbf{I}}$, the function $\mathcal{R}_\theta$ can be written as: $\mathcal{R}_\theta : x \in \mathbb{R}^{N_0} \mapsto \sum_{j=1}^{N_1} h_j(x) + \mathbf{b}_2$ where $h_j(x) = \mathbf{W}_2[:, j]\sigma(\mathbf{W}_1[j, :]x + \mathbf{b}_1[j])$ denotes the contribution of the $j$th hidden neuron. For hidden neurons corresponding to nonzero rows of $\mathbf{W}_1^k$, we can rescale the rows of $\mathbf{W}_1^k$, the columns of $\mathbf{W}_2^k$ and $\mathbf{b}_1^k$ such that the realization of $h_j$ is invariant. This is due to the fact that $\mathbf{w}_2\sigma(\mathbf{w}_1^\top x + b) = \|\mathbf{w}_1\|\mathbf{w}_2\sigma((\mathbf{w}_1/\|\mathbf{w}_1\|)^\top x + (b/\|\mathbf{w}_1\|))$ for any $\mathbf{w}_1 \neq \mathbf{0} \in \mathbb{R}^{N_0}, \mathbf{w}_2 \in \mathbb{R}^{N_2}, b \in \mathbb{R}$. Neurons corresponding to null rows of $\mathbf{W}_1^k$ are handled similarly,

in an iterative manner, by setting them to an arbitrary normalized row, setting the corresponding column of $\mathbf{W}_2^k$ to zero, and changing the bias $\mathbf{b}_2^k$ to keep the function $\mathcal{R}_\theta$ unchanged on $\mathbb{R}^{N_0}$, using that $\mathbf{w}_2\sigma(\mathbf{0}^\top x + b) + \mathbf{b}_2 = \mathbf{0}\sigma(\mathbf{v}^\top x + b) + (\mathbf{b}_2 + \mathbf{w}_2\sigma(b))$ for any normalized vector $\mathbf{v} \in \mathbb{R}^{N_0}$. Thus, we obtain $\theta'$ whose matrix of the first layer, $\mathbf{W}_1'$, has normalized rows and $\mathcal{R}_\theta = \mathcal{R}_{\theta'}$ on $\mathbb{R}^{N_0}$.

To construct $\tilde{\theta}$ with $\|\tilde{\mathbf{b}}_1\|_\infty \leq C$ we see that, by definition of $C$, if $\|\mathbf{w}_1\| = 1$ and $b \geq C$ then

$$\mathbf{w}_1^\top x + b \geq -C + b \geq 0, \qquad \forall x \in \Omega. \tag{20}$$

Thus, the function $\mathbf{w}_2\sigma(\mathbf{w}_1^\top x + b) = \mathbf{w}_2(\mathbf{w}_1^\top x + b)$ is linear on $\Omega$ and

$$\mathbf{w}_2\sigma(\mathbf{w}_1^\top x + b) + \mathbf{b}_2 = \mathbf{w}_2(\mathbf{w}_1^\top x + C) + ((b - C)\mathbf{w}_2 + \mathbf{b}_2)$$
$$= \mathbf{w}_2\sigma(\mathbf{w}_1^\top x + C) + ((b - C)\mathbf{w}_2 + \mathbf{b}_2)$$

Thus, for any hidden neuron with a bias exceeding $C$, the bias can be saturated to $C$ by changing accordingly the output bias $\mathbf{b}_2$, keeping the function $\mathcal{R}_\theta$ unchanged *on the bounded domain* $\Omega$ (but not on the whole space $\mathbb{R}^{N_0}$). Hidden neurons with a bias $b \leq -C$ can be similarly modified. Sequentially saturating each hidden neuron yields $\tilde{\theta}$ which satisfies all conditions of Lemma C.1. $\square$

**Lemma C.2.** *Consider $\Omega$ a bounded subset of $\mathbb{R}^{N_0}$, for any $\mathbf{I} = (I_2, I_1)$, given a continuous function $f \in \overline{\mathcal{F}_\mathbf{I}(\Omega)}$, there exists a sequence $\{\theta^k\}_{k \in \mathbb{N}}, \theta^k = (\mathbf{W}_i^k, \mathbf{b}_i^k)_{i=1}^2 \in \mathcal{N}_\mathbf{I}$ such that:*

*1. The sequence $\mathcal{R}_{\theta^k}$ admits $f$ as its uniform limit, i.e., $\lim_{k \to \infty} \|\mathcal{R}_{\theta^k} - f\|_\infty = 0$.*

*2. The sequence $\{(\mathbf{W}_1^k, \mathbf{b}_1^k)\}_{k \in \mathbb{N}}$ has a finite limit $(\mathbf{W}_1^\star, \mathbf{b}_1^\star)$ where $\mathbf{W}_1^\star$ has unit norm rows and $\mathrm{supp}(\mathbf{W}_1^\star) \subseteq I_1$.*

*Proof.* Given a function $f \in \overline{\mathcal{F}_\mathbf{I}(\Omega)}$, by definition, there exists a sequence $\{\theta^k\}_{k \in \mathbb{N}}, \theta^k \in \mathcal{N}_\mathbf{I}$ $\forall k \in \mathbb{N}$ such that $\lim_{k \to \infty} \|\mathcal{R}_{\theta^k} - f\|_\infty = 0$. We can assume that $\mathbf{W}_1^k$ has normalized rows and $\mathbf{b}_1^k$ is bounded using Lemma C.1. We can also assume WLOG that the parameters of the first layer (i.e $\mathbf{W}_1^k, \mathbf{b}_1^k$) have finite limits $\mathbf{W}_1^\star$ and $\mathbf{b}_1^\star$. Indeed, since both $\mathbf{W}_1^k$ and $\mathbf{b}_1^k$ are bounded (by construction from Lemma C.1), there exists a subsequence $\{\varphi_k\}_{k \in \mathbb{N}}$ such that $\mathbf{W}_1^{\varphi_k}$ and $\mathbf{b}_1^{\varphi_k}$ have finite limits and $\mathcal{R}_{\theta^{\varphi_k}} \to f$ as $\mathcal{R}_{\theta^k} \to f$. Replacing the sequence $\{\theta^k\}_{k \in \mathbb{N}}$ by $\{\theta^{\varphi_k}\}_{k \in \mathbb{N}}$ yields the desired sequence. Finally, since $\mathbf{W}_1^\star = \lim_{k \to \infty} \mathbf{W}_1^k$, $\mathbf{W}_1^\star$ obviously has normalized rows and $\mathrm{supp}(\mathbf{W}_1^\star) \subseteq I_1$. $\square$

**Definition C.1.** *Consider $\Omega$ bounded subset of $\mathbb{R}^d$, a function $f \in \overline{\mathcal{F}_\mathbf{I}(\Omega)}$ and a sequence $\{\theta^k\}_{k \in \mathbb{N}}$ as given by Lemma C.2. We define $(a_i, b_i) = (\mathbf{W}_1^\star[i, :], \mathbf{b}_1^\star[i])$ the limit parameters of the first layer corresponding to the $i$th neuron. We partition the set of neurons into two subsets (one of them may be empty):*

*1. Set of active neurons: $J := \{i \mid (\exists x \in \Omega, a_i x + b_i > 0) \wedge (\exists x \in \Omega, a_i x + b_i < 0)\}$.*

*2. Set of non-active neurons: $\bar{J} = [\![N_1]\!] \setminus J$.*

*For $i, j \in J$, we write $i \simeq j$ if $(\mathbf{W}_1^\star[j, :], \mathbf{b}_1^\star[j]) = \pm(\mathbf{W}_1^\star[i, :], \mathbf{b}_1^\star[i])$. The relation $\simeq$ is an equivalence relation.*

*We define $(J_\ell)_{\ell=1,\ldots,r}$ the equivalence classes induced by $\simeq$ and we use $(\alpha_\ell, \beta_\ell) := (a_i, b_i)$ for some $i \in J_\ell$ as the representative limit of the $\ell$th equivalence class. For $i \in J_\ell$, we have: $(a_i, b_i) = \epsilon_i(\alpha_\ell, \beta_\ell), \epsilon_i \in \{\pm 1\}$. We define $J_\ell^+ = \{i \in J_\ell \mid \epsilon_i = 1\} \neq \emptyset$ and $J_\ell^- = J_\ell \setminus J_\ell^+$.*

*For each equivalence class $J_\ell$, define $H_\ell := \{x \in \Omega \mid \alpha_\ell x + \beta_\ell = 0\}$ the boundary generated by neurons in $J_\ell$ and the positive (resp. negative) half-space partitioned by $H_\ell$, $H_\ell^+ := \{x \in \Omega \mid \alpha_\ell x + \beta_\ell > 0\}$ (resp. $H_\ell^- := \{x \in \Omega \mid \alpha_\ell x + \beta_\ell < 0\}$). For any $\epsilon > 0$ we also define the open half-spaces $H_\ell^{(\epsilon,+)} := \{x \in \mathbb{R}^d \mid \alpha_\ell^\top x + \beta_\ell > \epsilon\}$ and $H_\ell^{(\epsilon,-)} := \{x \in \mathbb{R}^d \mid \alpha_\ell^\top x + \beta_\ell < -\epsilon\}$.*

Definition C.1 groups neurons sharing the same "linear boundary" (or "singular hyperplane" as in [26]). This concept is related to "twin neurons" [28], which also groups neurons with the same active zone. This partition somehow allows us to treat classes independently. Observe also that

$$\mathrm{supp}(\alpha_\ell) \subseteq \bigcap_{i \in J_\ell} I_1[i, :], \forall 1 \leq \ell \leq r. \tag{21}$$

**Definition C.2** (Contribution of an equivalence class). *In the setting of Definition C.1, we define the contribution of the $i$th neuron $1 \leq i \leq N_1$ (resp. of the $\ell$th ($1 \leq \ell \leq r$) equivalence class) of $\theta^k$ as:*

$$h_i^k : \mathbb{R}^{N_0} \mapsto \mathbb{R}^{N_2} : x \mapsto \mathbf{W}_2^k[:, i]\sigma(\mathbf{W}_1^k[i, :]x + \mathbf{b}_1^k[i]) \,,$$

$$g_\ell^k : \mathbb{R}^{N_0} \mapsto \mathbb{R}^{N_2} : x \mapsto \sum_{i \in J_\ell} h_i^k(x) \,.$$

**Lemma C.3.** *Consider $\Omega = [-B, B]^d$, $f \in \overline{\mathcal{F}_\mathbf{I}(\Omega)}$ and a sequence $\{\theta^k\}_{k \in \mathbb{N}}$ as given by Lemma C.2, and $\alpha_\ell, \beta_\ell, 1 \leq \ell \leq r, \epsilon_i, i \in J$ as given by Definition C.1. There exist some $\gamma_\ell, \mathbf{b} \in \mathbb{R}^{N_2}, \mathbf{A} \in \mathbb{R}^{N_2 \times N_0}$ such that:*

$$f(x) = \sum_{i=1}^{r} \gamma_\ell \sigma(\alpha_\ell x + \beta_\ell) + \mathbf{A}x + \mathbf{b} \quad \forall x \in \Omega \tag{22}$$

$$\lim_{k \to \infty} \sum_{i \in J_\ell} \epsilon_i \mathbf{W}_2^k[:, i]\mathbf{W}_1^k[i, :] = \gamma_\ell \alpha_\ell, \quad \forall 1 \leq \ell \leq r \tag{23}$$

$$\lim_{k \to \infty} \sum_{i \in J_\ell} \epsilon_i \mathbf{b}_1^k[i]\mathbf{W}_2^k[:, i] = \gamma_\ell \beta_\ell, \quad \forall 1 \leq \ell \leq r \tag{24}$$

$$\mathtt{supp}(\gamma_\ell) \subseteq \bigcup_{i \in J_\ell} I_2[:, i], \quad \forall 1 \leq \ell \leq r \tag{25}$$

*Proof.* The proof is divided into three parts: We first show that there exist $\gamma_\ell, \mathbf{b} \in \mathbb{R}^{N_2}$ and $\mathbf{A} \in \mathbb{R}^{N_2 \times N_0}$ such that Equation (22) holds. The last two parts will be devoted to prove that equations (23) - (25) hold.

1. **Proof of Equation (22):** Our proof is based on a result of [26], which deals with the case of a scalar output (i.e, $N_2 = 1$). It is proved in [26, Theorem 3.8, Steps $3, 6, 7$] and states the following:

**Lemma C.4** (Analytical form of a limit function with scalar output [26]). *In case $N_2 = 1$ (i.e., output dimension equal to one), consider $\Omega = [-B, B]^d$, a scalar-valued function $f : \Omega \mapsto \mathbb{R}, f \in \overline{\mathcal{F}_\mathbf{I}(\Omega)}$ and a sequence as given by Lemma C.2, there exist $\mu \in \mathbb{R}^{N_0}, \gamma_\ell, \nu \in \mathbb{R}$ such that:*

$$f(x) = \sum_{\ell=1}^{r} \gamma_\ell \sigma(\alpha_\ell x + \beta_\ell) + \mu^\top x + \nu, \quad \forall x \in \Omega \tag{26}$$

Back to our proof, one can write $f = (f_1, \ldots, f_{N_2})$ where $f_j : \Omega \subseteq \mathbb{R}^{N_0} \mapsto \mathbb{R}$ is the function $f$ restricted to the $j$th coordinate. Clearly, $f_j$ is also a uniform limit on $\Omega$ of $\{\mathcal{R}_{\tilde{\theta}^k}\}_{k \in \mathbb{N}}$ for a sequence $\{\tilde{\theta}^k\}_{k \in \mathbb{N}}$ which shares the same $\mathbf{W}_1^k$ with $\{\theta^k\}_{k \in \mathbb{N}}$ but $\tilde{\mathbf{W}}_2^k$ is the $j$th row of $\mathbf{W}_2^k$. Therefore, $\{\tilde{\theta}^k\}_{k \in \mathbb{N}}$ also satisfies the assumptions of Lemma C.4, which gives us:

$$f_j(x) = \sum_{\ell=1}^{r} \gamma_{\ell,j} \sigma(\alpha_\ell x + \beta_\ell) + \mu_j^\top x + \nu_j, \quad \forall x \in \Omega$$

for some $\mu_j \in \mathbb{R}^{N_0}, \gamma_{i,j}, \nu_j \in \mathbb{R}$. Note that $\alpha_\ell, \beta_\ell$ and $r$ are not dependent on the index $j$ since they are defined directly from the considered sequence. Therefore, the function $f$ (which is the concatenation of $f_j$ coordinate by coordinate) is:

$$f(x) = \sum_{\ell=1}^{r} \gamma_\ell \sigma(\alpha_\ell x + \beta_\ell) + \mathbf{A}x + \mathbf{b}, \quad \forall x \in \Omega$$

with $\gamma_\ell = \begin{pmatrix} \gamma_{i,1} \\ \vdots \\ \gamma_{i,N_2} \end{pmatrix}, \mathbf{A} = \begin{pmatrix} \mu_1^\top \\ \vdots \\ \mu_{N_2}^\top \end{pmatrix}, \mathbf{b} = \begin{pmatrix} \nu_1 \\ \vdots \\ \nu_{N_2} \end{pmatrix}$.

2. **Proof for Equations** (23)-(24): With the construction of $\gamma$, we will prove Equation (23) and Equation (24). We consider an arbitrary $1 \leq \ell \leq r$. Denoting $\Omega^\circ$ the interior of $\Omega$ and $H_\ell := \{x \in \Omega \mid \alpha_\ell x + \beta_\ell = 0\}$ the hyperplane defined by the input weights and bias of the $\ell$-th class of

813 neurons, we take a point $x' \in (\Omega^\circ \cap H_\ell) \setminus \bigcup_{p \neq \ell} H_p$ and a fixed scalar $r > 0$ such that the open ball
814 $\mathcal{B}(x', r) \subseteq \Omega^\circ \setminus \bigcup_{p \neq \ell} H_p$. Notice that $x'$ is well-defined due to the definition of $J$ (Definition C.1).
815 In addition, $r$ also exists because $\Omega^\circ \setminus \bigcup_{p \neq \ell} H_p$ is an open set. Thus, there exists two constants
816 $0 < \delta < B$ and $\epsilon > 0$ such that:

(a) $\mathcal{B}(x', r) \subseteq [-(B-\delta), B-\delta]^d$.

(b) For each $p \neq \ell$, the ball $\mathcal{B}(x', r)$ is either included in the half-space $H_p^{(\epsilon,+)} := \{x \in \mathbb{R}^d \mid$
 $\alpha_p^\top x + \beta_p > \epsilon\}$ or in the half-space $H_p^{(\epsilon,-)} := \{x \in \mathbb{R}^d \mid \alpha_p^\top x + \beta_p < -\epsilon\}$.

(c) The intersection of $\mathcal{B}(x', r)$ with $H_\ell^{(\epsilon,+)}$ and $H_\ell^{(\epsilon,-)}$ are not empty.

821 For the remaining of the proof, we will use Lemma C.5, another result taken from [26]. We only state
822 the lemma. Its formal proof can be found in the proof of [26, Theorem 3.8, Steps 4-5].

823 **Lemma C.5** (Affine linear area [26]). *Given a sequence $\{\theta^k\}_{k \in \mathbb{N}}$ satisfying the second condition of*
824 *Lemma C.2, we have:*

825 *(a) For any $0 < \delta < B$, there exists a constant $\kappa_\delta$ such that $\forall i \in \bar{J}$, $h_i^k$ are affine linear on*
826 *$[-(B-\delta), B-\delta]^{N_0}$ for all $k \geq \kappa_\delta$.*
827 *(b) For any $\epsilon > 0$, there exists a constant $\kappa_\epsilon$ such that for each $1 \leq \ell \leq r$ and each $i \in J_\ell$ the*
828 *function $h_i^k$ is affine linear on $H_\ell^{(\epsilon,+)} \cup H_\ell^{(\epsilon,-)}$ for all $k \geq \kappa_\epsilon$.*

The lemma implies the existence of $K = \max(\kappa_\delta, \kappa_\epsilon)$ such that for all $k \geq K$, we have:

$$\sum_{p \neq \ell} g_p^k(x) = \mathbf{B}^k x + \nu^k, \qquad \forall x \in \mathcal{B}(x', r),$$

for some $\mathbf{B}^k \in \mathbb{R}^{N_2 \times N_0}, \nu^k \in \mathbb{R}^{N_2}$. Therefore, for $k \geq K$, we have:

$$\mathcal{R}_{\theta^k}(x) = \mathbf{B}^k x + \nu^k + \sum_{i \in J_\ell^+} \mathbf{W}_2^k[:, i](\mathbf{W}_1^k[i, :]x + \mathbf{b}_1^k[i]), \forall x \in \mathcal{B}(x', r) \cap H_\ell^{(\epsilon,+)}$$

$$\mathcal{R}_{\theta^k}(x) = \mathbf{B}^k x + \nu^k + \sum_{i \in J_\ell^-} \mathbf{W}_2^k[:, i](\mathbf{W}_1^k[i, :]x + \mathbf{b}_1^k[i]), \forall x \in \mathcal{B}(x', r) \cap H_\ell^{(\epsilon,-)}.$$

Since we proved that $f$ has the form Equation (22), there exist $\mathbf{C} \in \mathbb{R}^{N_2 \times N_0}, \mu \in \mathbb{R}^{N_2}$ such that

$$f(x) = (\mathbf{C} + \gamma_\ell \alpha_\ell)x + (\mu + \gamma_\ell \beta_\ell), \quad \forall x \in \mathcal{B}(x', r) \cap H_\ell^{(\epsilon,+)}$$
$$f(x) = \mathbf{C}x + \mu, \qquad \forall x \in \mathcal{B}(x', r) \cap H_\ell^{(\epsilon,-)}$$

829 As both $\mathcal{B}(x', r) \cap H_\ell^{(\epsilon,+)}$ and $\mathcal{B}(x', r) \cap H_\ell^{(\epsilon,-)}$ are open sets, and given our hypothesis of uniform
830 convergence of $\mathcal{R}_{\theta^k} \to f$, we obtain,

$$\lim_{k \to \infty} \mathbf{B}^k + \sum_{i \in J_\ell^+} \mathbf{W}_2^k[:, i]\mathbf{W}_1^k[i, :] = \mathbf{C} + \gamma_\ell \alpha_\ell$$

$$\lim_{k \to \infty} \mathbf{B}^k + \sum_{i \in J_\ell^-} \mathbf{W}_2^k[:, i]\mathbf{W}_1^k[i, :] = \mathbf{C}$$

$$\lim_{k \to \infty} \nu^k + \sum_{i \in J_\ell^+} \mathbf{b}_1^k[i]\mathbf{W}_2^k[:, i] = \mu + \gamma_\ell \beta_\ell \qquad (27)$$

$$\lim_{k \to \infty} \nu^k + \sum_{i \in J_\ell^-} \mathbf{b}_1^k[i]\mathbf{W}_2^k[:, i] = \mu.$$

831 Proof for Equation (27) can be found in Appendix C.4. Equations (23) and (24) follow directly from
832 Equation (27).

833 3. **Proof of Equation (25)**: Since $\alpha_\ell \neq 0$ (remember that $\|\alpha_\ell\| = 1$), this is an immediate conse-
834 quence of Equation (23) as each vector $\mathbf{W}_2^k[:, j], j \in J_\ell$ is supported in $I_2[:, j] \subseteq \cup_{i \in J_\ell} I_2[:, i]$. $\quad\square$

We state an immediate corollary of Lemma C.3, which characterizes the limit of the sequence of contributions $\{g_\ell^k\}_{k\in\mathbb{N}}$ of the $\ell$th equivalence class with $|J_\ell| = 1$.

**Corollary C.1.** *Consider $f \in \overline{\mathcal{F}_\mathbf{I}([-B,B]^d)}$ that admits the analytical form in Equation (22), a sequence $\{\theta^k\}_{k\in\mathbb{N}}$ as given by Lemma C.2, and Definition C.1. For all singleton equivalence classes $J_\ell = \{i\}, 1 \le \ell \le r$, we have $\lim_{k\to\infty} \mathbf{W}_2^k[:,i] = \gamma_\ell$ and $\lim_{k\to\infty} \|h_\ell^k - \gamma_\ell \sigma(\alpha_\ell^\top x + \beta_\ell)\|_\infty = 0$.*

*Proof.* We first prove that $\mathbf{W}_2^k[:,i]$ has a finite limit. In fact, applying the second point of Lemma C.3 for $J_\ell = \{i\}$, we have:
$$\lim_{k\to\infty} \mathbf{W}_2^k[:,i]\mathbf{W}_1^k[i,:] = \gamma_\ell \alpha_\ell$$
where $\gamma_\ell, \alpha_\ell$ are defined in Lemma C.3. Because $\lim_{k\to\infty} \mathbf{W}_1^k[i,:] = \alpha_\ell$ and $\|\alpha_\ell\|_2 = 1$, it follows that $\gamma_\ell = \lim_{k\to\infty} \mathbf{W}_2^k[:,i]$. To conclude, since we also have $\beta_\ell = \lim_{k\to\infty} \mathbf{b}_1^k[i]$, we obtain $h_\ell^k(\cdot) = \mathbf{W}_2^k[\ell,:]\sigma(\mathbf{W}_1^k[\ell,:] \cdot + \mathbf{b}_1^k[\ell]) \to \gamma_\ell \sigma(\alpha_\ell x + \beta_\ell)$ as claimed. $\qquad\square$

The nice thing about Corollary C.1 is that the contribution $g_\ell^k = h_\ell^k$ admits a (uniform) limit if $J_\ell = \{i\}$. Moreover, this limit is even implementable by using only the $i$th neuron because $\mathtt{supp}(\alpha_\ell) \subseteq I_1[i,:]$ and $\mathtt{supp}(\gamma_\ell) \subseteq I_2[:,i]$.

It would be tempting to believe that, for each $P \in \{\bar{J}\} \cup \{J_\ell \mid \ell = 1,\ldots,r\}$ the sequence of functions $\sum_{i\in P} g_i^k(x)$ must admit a limit (when $k$ tends to $\infty$) and that this limit is implementable using only neurons in $P$. This would obviously imply that $\mathcal{F}_\mathbf{I}(\Omega)$ is closed. This intuition is however wrong. For non-singleton equivalence class (i.e., for cases *not* covered by Corollary C.1), the limit function *does not necessarily exist* as we show in the following example.

**Example C.1.** *Consider the case where $\mathbf{N} = (1,3,1)$ and no support constraint, $\Omega = [-1,1]$, take the sequence $\{\theta^k\}_{k\in\mathbb{N}}$ which satisfies:*
$$\mathbf{W}_1^k = \begin{pmatrix} 1 \\ -1 \\ 1 \end{pmatrix}, \mathbf{b}_1^k = \begin{pmatrix} 0 \\ 0 \\ 1 \end{pmatrix}, \mathbf{W}_2^k = \begin{pmatrix} k & -k & -k \end{pmatrix}, \mathbf{b}_2^k = k$$

*Then for $x \in \Omega$, it is easy to verify that $\mathcal{R}_{\theta^k} = 0$. Indeed,*
$$\begin{aligned}
\mathcal{R}_{\theta^k}(x) &= \sum_{i=1}^{3} \mathbf{W}_2^k[:,i]\sigma(\mathbf{W}_1^k[i,:] + \mathbf{b}_1^k[i]) + \mathbf{b}_2^k \\
&= k\sigma(x) - k\sigma(-x) - k\sigma(x+1) + k \\
&= k(\sigma(x) - \sigma(-x)) - k(x+1) + k \quad (\text{since } x+1 \ge 0, \forall x \in \Omega) \\
&= kx - k(x+1) + k = 0
\end{aligned}$$

*Thus, this sequence converges (uniformly) to $f = 0$. Moreover, this sequence also satisfies the assumptions of Lemma C.2. Using the classification in Definition C.1, we have one class equivalence $J_1 = \{1,2\}$ and $\bar{J} = \{3\}$. The function $g_1^k(x) = k\sigma(x) - k\sigma(-x) = kx$, however, does not have any limit.*

### C.2.2    Actual proof of Theorem 4.2

Therefore, our analysis cannot treat each equivalence class entirely separately. The last result in this section is about a property of the matrix $\mathbf{A}$ in Equation (22). This is one of our key technical contributions in this work.

**Lemma C.6.** *Consider $\Omega = [-B,B]^d, f \in \overline{\mathcal{F}_\mathbf{I}(\Omega)}$ that admits the analytical form in Equation (22), a sequence $\{\theta^k\}_{k\in\mathbb{N}}$ as given by Lemma C.2, then the matrix $\mathbf{A} \in \overline{\mathcal{L}_{\mathbf{I}'}}$ where $\mathbf{I}' = (I_2[:,S], I_1[S,:]), S = \bar{J} \cup (\cup_{1 \le \ell \le r} J_\ell^-), \bar{J}, J_\ell^\pm$ are defined as in Definition C.1).*

Combining Lemma C.6 and the assumptions of Theorem 4.2, we can prove Theorem 4.2 immediately as follow:

*Proof of Theorem 4.2.* Consider $f \in \overline{\mathcal{F}_\mathbf{I}(\Omega)}$, we deduce that there exists a sequence of $\{\theta^k\}_{k\in\mathbb{N}}$ that satisfies the properties of Lemma C.2. This allows us to define $\bar{J}$ and equivalence classes

$J_\ell, 1 \leq \ell \leq r$ as well as $(\alpha_\ell, \beta_\ell)$ as in Definition C.1. Using Lemma C.3, we can also deduce an analytical formula for $f$ as in Equation (22):

$$f(x) = \sum_{i=1}^{r} \gamma_\ell \sigma(\alpha_\ell x + \beta_\ell) + \mathbf{A}x + \mathbf{b}, \quad \forall x \in \Omega.$$

Finally, Lemma C.6 states that matrix $\mathbf{A}$ in Equation (22) satisfies: $\mathbf{A} \in \overline{\mathcal{L}_{\mathbf{I}'}}$ with $\mathbf{I}' = (I_2[:, S], I_1[S, :]])$, where $S = \bar{J} \cup (\cup_{\ell=1}^{r} J_\ell^-)$. To prove that $f \in \mathcal{F}_{\mathbf{I}}$, we construct the parameters $\theta = \{(\mathbf{W}_i, \mathbf{b}_i)_{i=1}^2\}$ of the limit network as follows:

1. For each $1 \leq \ell \leq r$, choose one index $j \in J_\ell^+$ (which is possible since $J_\ell^+$ is non-empty). We set:

$$(\mathbf{W}_1[i, :], \mathbf{W}_2[:, i], \mathbf{b}_1[i]) = \begin{cases} (\alpha_\ell, \gamma_\ell, \beta_\ell) & \text{if } i = j \\ (\alpha_\ell, \mathbf{0}, \beta_\ell) & \text{otherwise} \end{cases} \tag{28}$$

This satisfies the support constraint because $\text{supp}(\alpha_\ell) \subseteq I_1[j, :]$ (by (21)) $\alpha_\ell = \lim_{k \to \infty} \mathbf{W}_1^k[j, :]$) and $I_2 = \mathbf{1}_{N_2 \times N_1}$. This is where we use the first assumption of Theorem 4.2. Without it, $\text{supp}(\gamma_\ell)$ might not be a subset of $I_2[:, j]$.

2. For $i \in S$: Since $\mathbf{A} \in \overline{\mathcal{L}_{\mathbf{I}'}}$ (cf Lemma C.6) and $\mathcal{L}_{\mathbf{I}'}$ is closed (second assumptions of Theorem 4.2), there exist two matrices $\hat{\mathbf{W}}_1, \hat{\mathbf{W}}_2$ such that: $\text{supp}(\hat{\mathbf{W}}_1) \subseteq I_1[:, S], \text{supp}(\hat{\mathbf{W}}_2) \subseteq I_2[S, :]$, and $\mathbf{A} = \hat{\mathbf{W}}_2 \hat{\mathbf{W}}_1$. We set:

$$(\mathbf{W}_1[i, :], \mathbf{W}_2[:, i], \mathbf{b}_1[i]) = (\hat{\mathbf{W}}_1[i, :], \hat{\mathbf{W}}_2[:, i], C) \tag{29}$$

where $C = \sup_{x \in \Omega} \|\hat{\mathbf{W}}_1 x\|_\infty$. This satisfies the support constraints $\mathbf{I}$ due to our choice of $\hat{\mathbf{W}}_1, \hat{\mathbf{W}}_2$. The choice of $C$ ensures that the function $h_i(x) := \mathbf{W}_2[i, :]\sigma(\mathbf{W}_1[i, :]x + \mathbf{b}_1[i])$ is *linear* on $\Omega$.

3. For $\mathbf{b}_2$: Let $\mathbf{b}_2 = \mathbf{b} - C\left(\sum_{i \in S} \hat{\mathbf{W}}_2[:, i]\right)$ ($\mathbf{b}$ is the bias in Equation (22)).

Verifying $\mathcal{R}_\theta = f$ on $\Omega$ is thus trivial since:

$$\mathcal{R}_\theta(x) = \sum_{i=1}^{N_1} \mathbf{W}_2[:, i]\sigma(\mathbf{W}_1[i, :]x + \mathbf{b}_1[i]) + \mathbf{b}_2$$

$$= \sum_{i \notin S} \mathbf{W}_2[:, i]\sigma(\mathbf{W}_1[i, :]x + \mathbf{b}_1[i]) + \sum_{i \in S} \mathbf{W}_2[:, i]\sigma(\mathbf{W}_1[i, :]x + \mathbf{b}_1[i]) + \mathbf{b}_2$$

$$= \sum_{\ell=1}^{r} \gamma_\ell \sigma(\alpha_\ell x + \beta_\ell) + \sum_{j \in S} \hat{\mathbf{W}}_2[:, i](\hat{\mathbf{W}}_1[i, :]x + C) + \mathbf{b} - C\left(\sum_{i \in S} \hat{\mathbf{W}}_2[:, i]\right)$$

$$= \sum_{\ell=1}^{r} \gamma_\ell \sigma(\alpha_\ell x + \beta_\ell) + \hat{\mathbf{W}}_2 \hat{\mathbf{W}}_1 x + \mathbf{b} = \sum_{\ell=1}^{r} \gamma_\ell \sigma(\alpha_\ell x + \beta_\ell) + \mathbf{A}x + \mathbf{b} = f. \quad \square$$

*Proof of Lemma C.6.* In this proof, we define $\Omega_\delta^\circ = (-B + \delta, B - \delta)^{N_0}, 0 < \delta < B$. The choice of $\delta$ is not important in this proof (any $0 < \delta < B$ will do).

The proof of this lemma revolves around the following idea: We will construct a sequence of functions $\{f^k\}_{k \in \mathbb{N}}$ such that, for $k$ large enough, $f^k$ has the following analytical form:

$$f^k(x) = \sum_{\ell=1}^{r} \gamma_\ell \sigma(\alpha_\ell x + \beta_\ell) + \mathbf{A}^k x + \mathbf{b}^k, \forall x \in \Omega_\delta^\circ \tag{30}$$

and $\lim_{k \to \infty} f^k(x) = f(x) \; \forall x \in \Omega \setminus (\cup_{\ell=1}^{r} H_\ell)$ (or equivalently $f^k$ converges pointwise to $f$ on $\Omega \setminus (\cup_{\ell=1}^{r} H_\ell)$) and $\mathbf{A}^k$ admits a factorization into two factors $\mathbf{A}^k = \mathbf{X}^k \mathbf{Y}^k$ satisfying $\text{supp}(\mathbf{X}^k) \subseteq I_2[:, S], \text{supp}(\mathbf{Y}^k) \subseteq I_1[S, :]$, so that $\mathbf{A}^k \in \mathcal{L}_{\mathbf{I}'}$. Comparing Equation (22) and Equation (30), we deduce that the sequence of affine functions $\mathbf{A}^k x + \mathbf{b}^k$ converges pointwise to the affine function $\mathbf{A}x + \mathbf{b}$ on the open set $\Omega_\delta^\circ \setminus (\cup_{\ell=1}^{r} H_\ell)$. Therefore, $\lim_{k \to \infty} \mathbf{A}^k = \mathbf{A}$ by Lemma C.7, hence the conclusion.

The rest of this proof is devoted to the construction of $f^k = \mathcal{R}_{\tilde{\theta}^k}$ where $\tilde{\theta}^k \in \mathcal{N}_{\mathbf{N}}$ are parameters of a neural network of the same dimension as those in $\mathcal{N}_{\mathbf{I}}$ but only *partially* satisfying the support constraint $\mathbf{I}$. To guarantee that $f^k$ converges pointwise to $f$, we construct $\tilde{\theta}^k$ based on $\theta^k$ and harness their relation.

**Choice of parameters.** We set $\tilde{\theta}^k = \{(\tilde{\mathbf{W}}_i^k, \tilde{\mathbf{b}}_i^k)_{i=1}^2\} \in \mathcal{N}_{\mathbf{N}}$ where $\tilde{\mathbf{W}}_2^k \in \mathbb{R}^{N_2 \times N_1}$, $\tilde{\mathbf{W}}_1^k \in \mathbb{R}^{N_1 \times N_0}$ are defined as follows, where we use $C^k := \sup_{x \in \Omega} \|\mathbf{W}_1^k x\|_\infty$:

- For inactive neurons $i \in \bar{J}$, we simply set $(\tilde{\mathbf{W}}_1^k[:,i], \tilde{\mathbf{W}}_2^k[i,:], \tilde{\mathbf{b}}_1^k[i]) = (\mathbf{W}_1^k[:,i], \mathbf{W}_2^k[i,:], \mathbf{b}_1^k[i])$.

- For each equivalence class of active neurons $1 \leq \ell \leq r$, we choose some $j_\ell \in J_\ell^+$ (note that $J_\ell^+$ is non-empty due to Definition C.1) and set the parameters $(\tilde{\mathbf{W}}_2^k[:,i], \tilde{\mathbf{W}}_1^k[i,:], \mathbf{b}_1[i]), i \in J_\ell$ as:

$$(\tilde{\mathbf{W}}_1^k[i,:], \tilde{\mathbf{W}}_2^k[:,i], \tilde{\mathbf{b}}_1^k[i]) = \begin{cases} (\mathbf{W}_1^k[i,:], \mathbf{W}_2^k[:,i], C^k), & \forall j \in J_\ell^- \\ (\mathbf{W}_1^k[i,:], \mathbf{0}, C^k), & \forall i \in J_\ell^+ \setminus \{j_\ell\} \\ (\alpha_\ell, \gamma_\ell, \beta_\ell), & i = j_\ell \end{cases} \tag{31}$$

For $i \in J_\ell \setminus \{j_\ell\}$, we clearly have: $\mathrm{supp}(\tilde{\mathbf{W}}_1^k[i,:]) \subseteq I_1[i,:]$ and $\mathrm{supp}(\tilde{\mathbf{W}}_2^k[:,i]) \subseteq I_2[:,i]$. The $j_\ell$-th column of $\tilde{\mathbf{W}}_2^k$ is the only one that does not necessarily satisfy the support constraint, as $\mathrm{supp}(\gamma_\ell) \nsubseteq I_2[:,j_\ell]$ in general.

- Finally, the output bias $\mathbf{b}_2^k$ is set as:

$$\tilde{\mathbf{b}}_2^k := \mathbf{b}_2^k + \sum_{\ell=1}^r \underbrace{\sum_{i \in J_\ell^-} (\mathbf{b}_1^k[i] - C^k)\mathbf{W}_2^k[:,i]}_{=:\xi_\ell^k} \tag{32}$$

**Proof that $f^k := R_{\tilde{\theta}^k}$ converges pointwise to $f$ on $\Omega \setminus (\cup_{\ell=1}^r H_\ell)$.** We introduce notations analog to Definition C.2: for every $x \in \mathbb{R}^{N_0}$ we define:

$$\tilde{h}_i^k(x) = \tilde{\mathbf{W}}_2^k[:,i]\sigma(\tilde{\mathbf{W}}_1^k[i,:]x + \tilde{\mathbf{b}}_1^k[i]), \quad i = 1, \ldots, N_1; \qquad \tilde{g}_\ell^k(x) = \sum_{i \in J_\ell} \tilde{h}_i^k(x), \quad \ell = 1, \ldots, r$$

By construction

$$\tilde{h}_i^k = h_i^k, \quad \forall i \in \bar{J}, \forall k, \tag{33}$$

and we further explicit the form of $\tilde{h}_i^k, i \in J_\ell$ for $x \in \Omega$ (but *not* on $\mathbb{R}^{N_0}$) as:

$$\tilde{h}_i^k(x) = \begin{cases} \mathbf{W}_2^k[:,i](\mathbf{W}_1^k[i,:]x + C^k), & \forall i \in J_\ell^- \\ 0, & \forall i \in J_\ell^+ \setminus \{j_\ell\} \\ \gamma_\ell \sigma(\alpha_\ell x + \beta_\ell), & i = j_\ell \end{cases}, \tag{34}$$

We justify our formula in Equation (34) as follow:

1. For $i \in J_\ell^-$: since $C^k = \sup_{x \in \Omega} \|\mathbf{W}_1^k x\|_\infty$ by construction, $\tilde{\mathbf{W}}_1^k[i,:]x + \mathbf{b}_1^k[i] = \mathbf{W}_1^k[i,:]x + \mathbf{b}_1^k[i] \geq 0$. The activation $\sigma$ acts simply as an identity function.

2. For $i \in J_\ell^+$: Because we choose $\tilde{\mathbf{W}}_2^k[:,i] = \mathbf{0}$.

3. For $i = j_\ell$: Obvious due to the construction in Equation (31).

Given $x \in \Omega \setminus (\cup_{\ell=1}^r H_\ell)$, we now prove that this construction ensures that for each $\ell \in \{1, \ldots, r\}$

$$\lim_{k \to \infty} (\tilde{g}_\ell^k(x) - g_\ell^k(x) + \xi_\ell^k) = 0. \tag{35}$$

This will imply the claimed poinwise convergence since

$$\lim_{k\to\infty} f^k(x) = \lim_{k\to\infty} R_{\tilde{\theta}^k}(x) = \lim_{k\to\infty}\left(\sum_{i\in\bar{J}}\tilde{h}_i^k(x) + \sum_{\ell=1}^r \tilde{g}_\ell^k(x) + \tilde{\mathbf{b}}_2^k\right)$$

$$\overset{(33)\&(35)}{=} \lim_{k\to\infty}\left(\sum_{i\in\bar{J}} h_i^k(x) + \sum_{\ell=1}^r g_\ell^k(x) - \sum_{\ell=1}^r \xi_\ell^k + \tilde{\mathbf{b}}_2^k\right)$$

$$\overset{(32)}{=} \lim_{k\to\infty}\left(\sum_{i\in\bar{J}} h_i^k(x) + \sum_{\ell=1}^r g_\ell^k(x) + \mathbf{b}_2^k\right) = \lim_{k\to\infty} R_{\theta^k}(x) = f(x).$$

906  To establish (35), observe that as $x \in \Omega \setminus (\cup_{\ell=1}^r H_\ell)$ we have $x \notin H_\ell$. We thus distinguish two cases:
907  **Case $x \in H_\ell^-$.**

908  Using (31) we show below that for $k$ large enough and $x \in H_\ell^-$, we have

$$\tilde{h}_i^k(x) - h_i^k(x) = \begin{cases} (C^k - \mathbf{b}_1^k[i])\mathbf{W}_2^k[:,i], & i \in J_\ell^- \\ 0, & i \in J_\ell^+ \end{cases} \tag{36}$$

and thus

$$\tilde{g}_\ell^k(x) - g_\ell^k(x) + \xi_\ell^k = \sum_{i\in J_\ell}\left(\tilde{h}_i^k(x) - h_i^k(x)\right) + \xi_\ell^k = \sum_{i\in J_\ell^-}(C^k - \mathbf{b}_1^k[i])\mathbf{W}_2^k[:,i] + \xi_\ell^k = 0.$$

909  We indeed obtain (36) as follows. Since $x \in H_\ell^-$, $\alpha_\ell x + \beta_\ell < 0$, i.e., $-\alpha_\ell x - \beta_\ell > 0$. Therefore,
910  given the definitions of $J_\ell^\pm$ (cf Definition C.1) we have:

- For $i \in J_\ell^-$: $\lim_{k\to\infty}(\mathbf{W}_1^k[i,:], \mathbf{b}_1^k[i]) = -(\alpha_\ell, \beta_\ell)$, hence for $k$ large enough, we have $\mathbf{W}_1^k[i,:]x + \mathbf{b}_1^k[i] > 0$ so that $\sigma(\mathbf{W}_1^k[i,:]x + \mathbf{b}_1^k[i]) = \mathbf{W}_1^k[i,:]x + \mathbf{b}_1^k[i]$ and, as expressed in (36):

$$\tilde{h}_i^k(x) - h_i^k(x) \overset{(34)}{=} \mathbf{W}_2^k[:,i](\mathbf{W}_1^k[i,:]x + C^k) - \mathbf{W}_2^k[:,i](\mathbf{W}_1^k[i,:]x + \mathbf{b}_1^k[i]) = (C^k - \mathbf{b}_1^k[i])\mathbf{W}_2^k[:,i].$$

911  - For $i \in J_\ell^+$: similarly, we have $\mathbf{W}_1^k[i,:]x + \mathbf{b}_1^k[i] < 0$ for $k$ large enough. Therefore, $h_i^k(x) = 0$
912  for $k$ large enough. The fact that we also have $\tilde{h}_i^k(x) = 0$ is immediate from Equation (34) if $i \neq j_\ell$,
913  and for $i = j_\ell$ we also get from Equation (34) that $\tilde{h}_i^k(x) = \gamma_\ell \sigma(\alpha_\ell x + \beta_\ell) = 0$ since $\alpha_\ell x + \beta_\ell < 0$.

914  **Case $x \in H_\ell^+$.** An analog to Equation (36) for $x \in H_\ell^+$ is

$$\tilde{h}_i^k(x) - h_i^k(x) = \begin{cases} \mathbf{W}_2^k[:,i](\mathbf{W}_1^k[i,:]x + C^k), & i \in J_\ell^- \\ -\mathbf{W}_2^k[:,i](\mathbf{W}_1^k[i,:]x + \mathbf{b}_1^k[i]), & i \in J_\ell^+ \setminus \{j\} \\ \gamma_\ell(\alpha_\ell x + \beta_\ell) - \mathbf{W}_2^k[:,i](\mathbf{W}_1^k[i,:]x + \mathbf{b}_1^k[i]), & i = j \end{cases} \tag{37}$$

915  We establish it before concluding for this case.

- For $i \in J_\ell^-$: by a reasoning analog to the case $x \in H_\ell^-$, we deduce that for $k$ large enough

$$\tilde{h}_i^k(x) - h_i^k(x) \overset{(34)}{=} \mathbf{W}_2^k[:,i](\mathbf{W}_1^k[i,:]x + C^k).$$

916  - For $i \in J_\ell^+$: a similar reasoning yields $h_i^k(x) = \mathbf{W}_2^k[:,i](\mathbf{W}_1^k[i,:]x + \mathbf{b}_1^k[i])$ for $k$ large enough,
917  while Equation (34) yields $\tilde{h}_{j_\ell}^k(x) = \gamma_\ell \sigma(\alpha_\ell x + \beta_\ell) = \gamma_\ell(\alpha_\ell x + \beta_\ell)$ (since $\alpha_\ell x + \beta_\ell > 0$ as $x \in H_\ell^+$)
918  and $\tilde{h}_i^k(x) = 0$ if $i \neq j_\ell$.

Using (37) we obtain for $k$ large enough

$$
\begin{aligned}
\tilde{g}_\ell^k(x) - g_\ell^k(x) + \xi_\ell^k &= \sum_{i \in J_\ell} \left( \tilde{h}_i^k(x) - h_i^k(x) \right) + \xi_\ell^k \\
&= \sum_{i \in J_\ell^-} \mathbf{W}_2^k[:,i](\mathbf{W}_1^k[i,:]x + C^k) - \sum_{i \in J_\ell^+} \mathbf{W}_2^k[:,i](\mathbf{W}_1^k[i,:]x + \mathbf{b}_1^k[i]) + \gamma_\ell(\alpha_\ell x + \beta_\ell) + \xi_\ell^k \\
&\overset{(32)}{=} \Big( \sum_{i \in J_\ell^-} \mathbf{W}_2^k[:,i]\mathbf{W}_1^k[i,:] - \sum_{j \in J_\ell^+} \mathbf{W}_2^k[:,i]\mathbf{W}_1^k[i,:] + \gamma_\ell \alpha_\ell \Big) x \\
&\quad + \Big( \underbrace{\xi_\ell^k + \sum_{i \in J_\ell^-} \mathbf{W}_2^k[:,i]C^k - \sum_{i \in J_\ell^+} \mathbf{W}_2^k[:,i]\mathbf{b}_1^k[i] + \gamma_\ell \beta_\ell}_{\sum_{i \in J_\ell^-} \mathbf{W}_2^k[:,i]\mathbf{b}_1^k[i]} \Big)
\end{aligned}
$$

where in the last line we used the expression of $\xi_\ell^k$ from (32). Due to Equations (23) and (24) it follows that $\lim_{k \to \infty} \tilde{g}_\ell^k(x) - g_\ell^k(x) + \xi_\ell^k = 0, \forall x \in H_\ell^+$.

Thus combining both cases, we conclude that $\lim_{k \to \infty} \tilde{g}_\ell^k(x) - g_\ell^k(x) + \xi_\ell^k = 0, \forall x \notin H_\ell$, as desired.

**Proof of the expression (30) with $\mathbf{A}^k \in \mathcal{L}_{\mathbf{I}'}$ for large enough $k$.** From (34), we first deduce that

$$
f^k(x) = \sum_{i=1}^{N_1} \tilde{h}_i^k(x) + \tilde{\mathbf{b}}_2^k = \sum_{\ell=1}^{r} \gamma_\ell \sigma(\alpha_\ell x + \beta_\ell) + \sum_{i \in S} \tilde{h}_i^k(x) + \tilde{\mathbf{b}}_2^k, \quad \forall x \in \mathbb{R}^{N_0}.
$$

where we recall that $S := \bar{J} \cup (\cup_{1 \le \ell \le r} J_\ell^-)$. There only remains to show that, for $k$ large enough, we have $\sum_{i \in S} \tilde{h}_i^k(x) = \mathbf{A}^k x + \mathbf{b}^k$ for every $x$ *in the restricted domain* $\Omega_\delta^\circ$, where $\mathbf{A}^k \in \mathcal{L}_{\mathbf{I}'}$ and $\mathbf{b}^k \in \mathbb{R}^{N_2}$. Note that for $i \in J_\ell$, our construction assures that $\tilde{h}_i^k$ is affine on $\Omega$. Moreover, in the restricted domain $\Omega_\delta^\circ$, for $k \ge \kappa_\delta$ large enough, $\tilde{h}_i^k, i \in \bar{J}$ also behave like affine functions (cf Lemma C.5). Therefore,

$$
\sum_{i \in S} \tilde{h}_i^k(x) = \Big( \sum_{i \in S} \delta_i^k \tilde{\mathbf{W}}_2^k[:,i]\tilde{\mathbf{W}}_1^k[i,:] \Big) x + \mathbf{c}^k, \quad \forall x \in \Omega_\delta^\circ, k \ge \kappa_\delta
$$

for some vector $\mathbf{c}^k$ and binary scalars $\delta_i^k$. In fact, $\delta_i^k = 0$ if $i \in \bar{J}^- := \{j \in \bar{J} \mid \mathbf{W}_1^\star[j,:]x + \mathbf{b}_1^\star[j] \le 0, \forall x \in \Omega\}$ and $\delta_k^i = 1$ otherwise. Thus, one chooses $\mathbf{A}^k = \sum_{i \in S} \delta_i^k \tilde{\mathbf{W}}_2^k[:,i]\tilde{\mathbf{W}}_1^k[i,:], \mathbf{b}^k = \mathbf{c}^k$ and the construction is complete. This construction allows us to write $\mathbf{A}^k = \hat{\mathbf{W}}_2^k \hat{\mathbf{W}}_1^k$ with:

$$
\begin{aligned}
\hat{\mathbf{W}}_1^k &= \tilde{\mathbf{W}}_1^k[S,:] \\
\hat{\mathbf{W}}_2^k &= \tilde{\mathbf{W}}_2^k[:,S]\texttt{diag}(\{\nu_i^k \mid i = 1, \ldots, N_1\})
\end{aligned}
$$

where $\texttt{diag}(\{\nu_i^k \mid i = 1, \ldots, N_1\}) \in \mathbb{R}^{N_1 \times N_1}$ is a diagonal matrix, $\nu_i^k = \delta_i^k$ for $i \in S$ and 0 otherwise. It is also evident that $\texttt{supp}(\hat{\mathbf{W}}_2^k[:,S]) \subseteq I_2[:,S], \texttt{supp}(\hat{\mathbf{W}}_1^k[S,:]) \subseteq I_1[S,:]$. (since the multiplication with a diagonal matrix does not increase the support of a matrix). This concludes the proof. $\qquad\square$

## C.3 Proof for Corollary 4.2

*Proof.* The proof is inductive on the number of hidden neurons $N_1$:

1. Basic case $N_1 = 1$: Consider $\theta := \{(\mathbf{W}_i, \mathbf{b}_i)_{i=1}^2\} \in \mathcal{N}_\mathbf{I}$, the function $\mathcal{R}_\theta$ has the form:

$$
\mathcal{R}_\theta(x) = \mathbf{w}_2 \sigma(\mathbf{w}_1^\top x + \mathbf{b}_1) + \mathbf{b}_2
$$

where $\mathbf{w}_1 = \mathbf{W}_1[1,:] \in \mathbb{R}^{N_0}, \mathbf{w}_2 = \mathbf{W}_2[1,1] \in \mathbb{R}$. There are two possibilities:

   (a) $I_2 = \emptyset$: then $\mathbf{w}_2 = 0$, $\mathcal{F}_I$ is simply a set of constant functions on $\Omega$, which is closed.

(b) $I_2 = \{(1,1)\}$: We have $I_2 = \mathbf{1}_{1 \times N_1}$, which makes the first assumption of Theorem 4.2 satisfied. To check that the second assumption of Theorem 4.2 also holds, we consider all the possible non-empty subsets $S$ of $[\![1]\!]$: there is only one non-empty subset of $I_2$, which is $S = [\![1]\!]$. In that case, $\mathcal{L}_{\mathbf{I}_S} = \{\mathbf{W} \in \mathbb{R}^{1 \times N_0} \mid \text{supp}(\mathbf{W}) \subseteq I_1\}$, which is closed (since $\mathcal{L}_{\mathbf{I}_S}$ is isomorphic to $\mathbb{R}^{|I_1|}$). The result thus follows using Theorem 4.2.

2. Assume the conclusion of the theorem holds for all $1 \leq N_1 \leq k$ (and any $N_0 \geq 1$). We need to prove the result for $N_1 = k + 1$. Define $H = \{i \mid I_2[1, i] = 1\}$ the set of hidden neurons that are allowed to be connected to the output via a nonzero weight. Consider two cases:

(a) If $|H| \leq k$, we have $\mathcal{F}_{\mathbf{I}} = \mathcal{F}_{\mathbf{I}_H}$, which is closed due to the induction hypothesis.
(b) If $H = [\![k+1]\!]$, we can apply Theorem 4.2. Indeed, since $I_2 = \mathbf{1}_{1 \times N_1}$, the first condition of Theorem 4.2 is satisfied. In addition, for any non-empty $S \subseteq [\![N_1]\!]$, define $\mathcal{H} := \cup_{i \in S} I[i, :] \subseteq [\![N_0]\!]$ the union of row supports of $I_1[S, :]$. It is easy to verify that $\mathcal{L}_{\mathbf{I}_S}$ is isomorphic to $\mathbb{R}^{|\mathcal{H}|}$, which is closed. As such, Theorem 4.2 can be applied. $\qquad\square$

## C.4 Other technical lemmas

**Lemma C.7** (Convergence of affine function). *Let $\Omega$ be a non-empty interior subset $\mathbb{R}^n$. If the sequence $\{f^k\}_{k \in \mathbb{N}}, f^k : \mathbb{R}^n \mapsto \mathbb{R}^m : x \mapsto \mathbf{A}^k x + \mathbf{b}^k$ where $\mathbf{A}^k \in \mathbb{R}^{m \times n}, \mathbf{b}^k \in \mathbb{R}^m$ converges pointwise to a function $f$ on $\Omega$, then $f$ is affine (i.e., $f = \mathbf{A}x + \mathbf{b}$ for some $\mathbf{A} \in \mathbb{R}^{m \times n}, \mathbf{b} \in \mathbb{R}^m$). Moreover, $\lim_{k \to \infty} \mathbf{A}^k = \mathbf{A}$ and $\lim_{k \to \infty} \mathbf{b}_k = \mathbf{b}$.*

*Proof.* Consider $x_0 \in \Omega'$, an open subset of $\Omega$ ($\Omega'$ exists since $\Omega$ is a non-empty interior subset of $\mathbb{R}^n$). Define $g^k(x) = f^k(x) - f^k(x_0)$ and $g(x) = f(x) - g(x_0)$. The function $g^k$ is linear and $g^k$ converges pointwise to $g$ on $\Omega$ (and thus, on $\Omega'$). We first prove that $g$ is linear. Indeed, for any $x, y \in \Omega, \alpha, \beta \in \mathbb{R}$ such that $\alpha x + \beta y \in \Omega$, we have:

$$
\begin{aligned}
g(\alpha x + \beta y) &= \lim_{k \to \infty} g^k(\alpha x + \beta y) \\
&= \lim_{k \to \infty} \alpha g^k(x) + \beta g^k(y) \\
&= \alpha \lim_{k \to \infty} g^k(x) + \beta \lim_{k \to \infty} g^k(y) \\
&= \alpha g(x) + \beta g(y)
\end{aligned}
$$

Therefore, there must exist $\mathbf{A} \in \mathbb{R}^{m \times n}$ such that $g(x) = \mathbf{A}x$. Choosing $\mathbf{b} := g(x_0)$, we have $f(x) = g(x) + g(x_0) = \mathbf{A}x + \mathbf{b}$.

Moreover, since $\Omega'$ is open, there exists a positive $r$ such that the ball $\mathcal{B}(x, r) \subseteq \Omega'$. Choosing $x_i = x_0 + (r/2)\mathbf{e}_i$ with $\mathbf{e}_i$ the $i$th canonical vector, we have:

$$
\lim_{k \to \infty} g^k(x_i) = \lim_{k \to \infty} (r/2)\mathbf{A}^k \mathbf{e}_i = (r/2)\mathbf{A}e_i,
$$

or, equivalently, the $i$th column of $\mathbf{A}$ is the limit of the sequence generated by the $i$th column of $\mathbf{A}^k$. Repeating this argument for all $1 \leq i \leq n$, we have $\lim_{k \to \infty} \mathbf{A}^k = \mathbf{A}$. This also implies $\lim_{k \to \infty} \mathbf{b}^k = \mathbf{b}$ immediately. $\qquad\square$

# D Closedness does not imply the best approximation property

Since we couldn't find any source discussing the fact that closedness does not imply the BAP, we provide an example to show this fact.

Consider $C^0([-1,1])$ the set of continuous functions on the interval $[-1,1]$, equipped with the norm sup $\|f\|_\infty = \max_{x \in [-1,1]} |f(x)|$, and define $S$, the subset of all functions $f \in C^0([-1,1])$ such that:

$$
\int_0^1 f \, dx - \int_{-1}^0 f \, dx = 1
$$

It is easy to verify that $S$ is closed. We show that the constant function $f = 0$ does not have a projection in $S$ (i.e., a function $g \in S$ such that $\|f - g\|_\infty = \inf_{h \in S} \|f - h\|_\infty$).

First we observe that since $f = 0$, we have $\|f - h\|_\infty = \|h\|_\infty$ for each $h \in S$, and we show that $\inf_{h \in S} \|f - h\|_\infty \geq 1/2$. Indeed, for $h \in S$ we have:

$$1 = \int_0^1 h \, dx - \int_{-1}^0 h \, dx \leq \left| \int_0^1 h \, dx \right| + \left| \int_{-1}^0 h \, dx \right| \leq 2\|h\|_\infty = 2\|f - h\|_\infty. \qquad (38)$$

Secondly, we show a sequence of $\{h_n\}_{k \in \mathbb{N}}$ such that $h_n \in S$ and $\lim_{n \to \infty} \|h_n\|_\infty = 1/2$. Consider the odd function $h_n$ (i.e. $h_n(x) = -h_n(-x)$) such that:

$$h_n(x) = \begin{cases} c_n, & x \in [1/n, 1] \\ n c_n x & x \in [0, 1/n) \end{cases}$$

where $c_n = n/(2n - 1)$. It is evident that $h_n \in S$ because:

$$\int_0^1 h_n \, dx - \int_{-1}^0 h_n \, dx = 2 \int_0^1 h_n \, dx = 2 \left( \int_0^{1/n} h_n \, dx + \int_{1/n}^1 h_n \, dx \right)$$

$$= 2 \left( \frac{c_n}{2n} + \frac{c_n(n-1)}{n} \right) = \frac{c_n(2n-1)}{n} = 1$$

Moreover, we also have $\lim_{n \to \infty} \|h_n\|_\infty = \lim_{n \to \infty} c_n = 1/2$.

Finally, we show that $1/2$ cannot be attained. By contradiction, assume that there exists $g \in S$ such that $\|f - g\|_\infty = 1/2$, i.e., as we have seen, $\|g\|_\infty = 1/2$. Using Equation (38), the equality will only hold if $g(x) = 1/2$ in $[0, 1]$ and $g(x) = -1/2$ in $[-1, 0]$. However, $g$ is not continuous, a contradiction.

