# OpenReview forum: "Does a sparse ReLU network training problem always admit an optimum ?"
_NeurIPS.cc/2023/Conference — NeurIPS 2023 poster_

### Official Review · Reviewer_ay34 · 2023-06-22

**Soundness:** 4 excellent
**Presentation:** 3 good
**Contribution:** 3 good
**Rating:** 7
**Confidence:** 4

**Summary:**

The authors are interested in the following question : « Given a deep learning architecture possibly with sparsity constraints, does its corresponding optimisation problem actually admit an optimal $\theta^*$? » While it is mostly taken for granted that the answer is « yes », the author proves it to be true in some cases, and false in others. They derive an algorithm in order to verify a sufficient condition in order for there to exist. More then a single way the sparseness of the considered network in ensured is considered in the article.

**Strengths:**

1. The relevance of the work is well defended in the introduction.

2. The results concern a large pool of predictors, considering a regression task.

3. The results concern more than one way of ensuring networks to be sparse.

4. The results seem to be a significant add-on to the literature concerning closeness and non-closeness of problems involving the training of (sometimes sparse) neural networks.

5. Example 3.1., though having some flaws (see Weaknesses - Major - 1.2.), is really important for illustrating what might otherwise be a bit hard to grasp for the reader, and proving the issues that might be encountered when not considering the initial question : « Given a deep learning architecture possibly with sparsity constraints, does its corresponding optimisation problem actually admit an optimal $\theta^*$? ».

**Weaknesses:**

_Major_

1.1. Regularisation is indeed (lines 41-42) a common way to bypass this question, but the notion of regularisation in itself is, especially when it comes to neural networks, mostly important for limiting over-fitting. I didn’t find the arguments regarding the discarding of regularisation for neural networks training really compelling (lines 44-47), especially since it wasn’t supported by any work from the literature.

1.2. Concerning Example 3.1 and the discussion that follows (lines 216-221), the discussion is based on what's displayed on Figure 1 only, forgetting about validation / test loss; saying that $L^2$ regularisation « might be detrimental » thus forget about validation / test loss. And if the goal is simply to over-fit a problem, then even smallest regularisation would make the parameter converge to certain values, practically not affecting the obtained error in train, which probably wasn't the case, since a single (and arbitrary) regularisation parameter was applied for each training.

2. The assumptions required for having Proposition 3.1. to hold (continuous, coercive) are limiting, making the use of 0-1 loss impossible (and cross-entropy as well, if I’m correct). The way the networks are defined too, since no soft-max activation function can be used on the output layer. Those loss functions (and the soft-max) being crucial when it comes to classifications, the results are mostly relevant only for regression, which is really not the most popular type of task when it comes to neural networks. Though it has been briefly mentioned on lines 176-177, I feel like the authors were not honest enough regarding how limiting the assumptions and the way the networks are defined are. For example, a sentence such as « Though our study theoretically applies to classification schemes, it more naturally suites regression schemes, considering the assumptions made in [...] » in the abstract, or at least in the Introduction section would be sufficient.

3. The motivations of the work are partially theoretical and mostly practical, but the results, though interesting, for it proves not every network has an optimal set of parameters in different settings, are mostly inapplicable in practice, limiting the impact of the work.

4. The terminology in Table 1 might be misleading, since the term « shallow » is vague and that when it comes to Theorem 3.4 and Corollaries 3.1, 4.1, 4.2 and 4.3, « shallow » stands for « 1-hidden-layer ». What are the limitations for the works from the literature with regards to the depth of the networks? If it is 1-hidden-layer, then the whole table should refer to « 1-hidden-layers architecture », and not « shallow architecture »; if it is deeper than 1-hidden-layer in at least one case, than the underlying maximum depth should be displayed in the table.



_Typos / Minor_

1. Line 126 : « agrees with $\mathbf{A}$ on rows in $S_{\underline{r}}$ (resp. columns in in $S_{\underline{c}}$) »

2. I expect Lemma 3.2. of the main paper to be coherent with what is stated on lines 505-506 for a camera-ready version of the article.

3. Lines 259 and 260 : The statement « This constraint on the sparsity level of each 260 layer is widely used in many works on sparse neural networks » is not supported by any citation.

4. Line 261 : « Consider scalar-val\underline{u}ed ».

5. Line 359 : « cf\underline{.} »

6. Using $\Omega$ for both the finite and the non-close cases is a bit confusing; it might be more convenient to differentiate these two cases with a separate notation.


**Questions:**

1. If the practical interest in studying problems with finite $\Omega$ is clear, I have a hard time understanding the interest of studying non-closed $\Omega$ and how it applies to real-world situations.

**Limitations:**

I feel like the authors downplayed the importance of some limitations of their work (see Weaknesses – Major).

---

> ### Author Rebuttal · Authors · 2023-08-06
>
> We thank the referee for his/her comments and questions.
>
> **Comments on the weakness**
>
> 1.1. While regularization with a coercive function indeed ensures the existence of a minimizer, this however implies tradeoffs between the data fitting term and the penalty, usually tuned via a parameter $\lambda$.
> Unfortunately, there may not be any satisfying tradeoff in non-closed cases, as is illustrated in Figure 1 (in the pdf of the global response) under the setting of Example 3.1 (in our submitted paper). Four curves display the concrete tradeoffs obtained along the training trajectory, when optimizing a regularized loss with four values of $\lambda$. In addition, we display random (yellow) and optimum (black) "oracle" tradeoffs obtained using approximate LU factorizations of the matrix $A$ of Example 3.1. These oracles are built as (exact) LU factorizations of perturbed matrices $A+N$, where $N$ is a noise matrix scaled to achieve prescribed approximation levels (and therefore specified empirical losses). Even though arbitrarily small empirical loss is theoretically possible, reaching an empirical loss below $0.1$ with an oracle approximate LU factorization requires LU factors with a norm of the order of $10^3$. The required norm quickly increases when improving the precision of the oracle LU approximation.
>
> 1.2. We thank the referee for this remark.  The validation error is indeed important. Choosing to display the Jacobian loss was motivated by the fact that it is indeed essentially proportional to the validation loss, as illustrated in Figure 2 in the pdf of the global response (with more values of $\lambda$ than in Figure 1 in the submitted paper, as a sanity check). This can also be proved theoretically under some assumptions. This was however too implicit and we will replace the Jacobian loss in Figure 1 (of the submitted paper) by the validation loss.
>
> 2. The assumptions made in Proposition 3.1 are indeed natural for the regression case but not for the classification case, and we agree with the referee that it is worth making the remark more visible since the beginning of the paper, this will be fixed in the final version. In the classification case, using the soft-max after the last layer together with the cross-entropy loss function indeed leads to an optimization problem with no optimum (regardless of the architecture) when given a *single* training pair $(x,y)$. This is due to the fact that changing either the bias or the scales of the last layer can lead the output of the soft-max arbitrarily close to an ideal Dirac mass. It is an interesting challenge to identify whether sufficiently many and diverse training samples (as in concrete learning scenarios) make the problem better posed, and amenable to a relevant closedness analysis. This point will be highlighted after Proposition 3.1.
>
> 3. The main contribution of the paper is to shed light on a phenomenon (absence of an optimum) that appears even in the simplest MLP architecture, which was mostly overlooked before (with the exception of few specialized papers, presenting results based on stronger assumptions). The first concrete consequence of our work is the possibility of detecting if a pruned MLP support poses a potential problem. A more general study of closedness for other architectures is left for future work, as this would require different tools. For MLPs, indeed the analysis is based on known results of matrix factorization. It seems possible for instance to generalize this analysis to the case of skip connections, because networks with skip connections can be modeled as larger ReLU networks with blocks of identity matrices as weights, so that their presence would add a further linear term in the Equation (27) (Lemma C.4, Appendix), which could be treated in the same way as the one that is already there. Convolutive layers are also likely to be amenable to an adapted analysis.
>
> 4. We will consistently use "one-hidden-layer".
>
> **Answer for question**
>
> We agree that the case of a finite $\Omega$ is the one encountered in practice when learning from actual training sets. Considering a domain $\Omega$ such as the unit cube, with non-empty interior, is indeed mostly of theoretical interest, but seems, for example, crucial when analyzing generalization properties, as the "optimal target function" is traditionally expressed as the minimizer of an expected risk defined as an integral on such domains or even on the whole space. Understanding whether this optimum target function exists is thus of interest.

---

> > ### Comment · Reviewer_ay34 · 2023-08-14
> >
> > I thank the authors for their insightful remarks. I'm glad to see a few changes will be made in the final version of the manuscript, reflecting the comments of the various reviewers. I stand by my score and recommend the acceptance of the paper.

---

### Official Review · Reviewer_oxZZ · 2023-07-01

**Soundness:** 4 excellent
**Presentation:** 3 good
**Contribution:** 3 good
**Rating:** 6
**Confidence:** 3

**Summary:**

This work studies the existence of global minimum in a sparse deep learning setting

**Strengths:**

I think this work touches on a very important problem that is understudied in the field: the existence of the global minimum for deep nonlinear networks. It is easy to image toy examples for which the global minimum does not exist, but until this work, there does not exist a formal study of this problem

The problem is important and novel. Linking this problem to training a sparse network is novel, I thus support its publication, though with some reservation

**Weaknesses:**

To me, I think the main problem is that the results are quite weak, and that the main example is not quite convincing or relevant. The example of a LU network is quite artificial and it is hard to imagine that this situation arises in deep learning. I think this work can benefit greatly if it identifies a more relevant and convincing example (and it should not be difficult to achieve). This is the main reason I give this work a weak accept

There is one minor problem that I do not find serious but is worth some attention. While the paper mostly motivates from the viewpoint of the popular pruning literature, this work can benefit from discussing more about regularization-based methods for compressing neural networks. For example, see the recent work in https://arxiv.org/abs/2210.01212, which also discusses the existence of the global minimum. At its face value, the results in this work seems to motivate and advocate the use of regularization-based compression methods in deep learning; is this interpretation correct? If not, why? I think discussing this point can better clarify the implication of this work

**Questions:**

See weakness

**Limitations:**

I think the work discussed the limitations well

---

> ### Author Rebuttal · Authors · 2023-08-06
>
> We thank the referee for his/her comments.
>
> **Comments on the weakness**
>
> For the *first* point, our Example 3.1 about the "LU architecture" is meant to be pedagogical, and its ultimate goal is to show that even in pretty simple cases there may be actual issues related to non-closedness. In our response to Reviewer 5AbX (https://openreview.net/forum?id=dTj5tH94xv&noteId=ozVHRnjQBI), we highlight that our results can also be used more concretely to detect supports leading to a non-closedness problem and that this occurs frequently for randomly pruned MLPs which serve as a baseline for sparse DNNs training [1].
>
> For the *second* point,  our main message is not to advocate the use of regularization-based compression, but rather to suggest that it is worth detecting supports leading to non-closedness.
>
> While regularization with a coercive function indeed ensures the existence of a minimizer, this however implies tradeoffs between the data fitting term and the penalty, usually tuned via a parameter $\lambda$.
> Unfortunately, there may not be any satisfying tradeoff in non-closed cases, as is illustrated in Figure 1 (in the pdf of the global response) under the setting of Example 3.1 (in our submitted paper). Four curves display the concrete tradeoffs obtained along the training trajectory, when optimizing a regularized loss with four values of $\lambda$. In addition, we display random (yellow) and optimum (black) "oracle" tradeoffs obtained using approximate LU factorizations of the matrix $A$ of Example 3.1. These oracles are built as (exact) LU factorizations of perturbed matrices $A+N$, where $N$ is a noise matrix scaled to achieve prescribed approximation levels (and therefore specified empirical losses). Even though arbitrarily small empirical loss is theoretically possible, reaching an empirical loss below $0.1$ with an oracle approximate LU factorization requires LU factors with a norm of the order of $10^3$. The required norm quickly increases when improving the precision of the oracle LU approximation.
>
> **References**
>
> [1] S. Liu, T. Chen, X. Chen, L. Shen, D.-C. Mocanu, Z. Wang, M. Pechenizkiy, The Unreasonable Effectiveness of Random Pruning: Return of the Most Naive Baseline for Sparse Training, International Conference on Learning Representations, ICLR, 2022.

---

> > ### Comment · Reviewer_oxZZ · 2023-08-16
> > **reply**
> >
> > Thanks for the response. As there is no significant concern, I keep my original score of weak acceptance.

---

### Official Review · Reviewer_7u6K · 2023-07-01

**Soundness:** 4 excellent
**Presentation:** 3 good
**Contribution:** 3 good
**Rating:** 7
**Confidence:** 3

**Summary:**

This paper provides necessary and sufficient conditions for whether a neural network training problem admits an optimal solution, that is, whether weights and biases exist such that the infimum of the loss function is actually attained. This is done for the classical case of empirical risk minimization on a finite training set as well as for function approximation on a continuous input domain.

**Strengths:**

- I think the investigated question is highly interesting already for purely theoretical reasons: the existence of an optimal solution is such a fundamental property of an optimization problem. We should be able to answer this question for our "favorite" optimization problem, namely training a neural network. This work seems to play a crucial role in (i) posing this question and (ii) providing results for some cases.
- Apparently the non-existence of an optimal solution can also explain divergence in practical settings, which I find compelling. Still, I would like to emphasize that the paper is a theory paper and should be judged as such.
- The paper involves quite some amount of non-trivial mathematics. Although the tight reviewing schedule does not allow me to verify all details, the parts I read seem to be mathematically sound.

**Weaknesses:**

- Many cases for the posed question remain open. In particular, most of the results are concerned with 2-layer NNs only.
- The presentation of the results in the intro could be improved. Some definitions could be made earlier (or at all). See my detailed comments below. Also I find it quite difficult to read the Tables 1 and 2. One basically has to read the rest of the paper in order to truely understand what these tables are about.

Comments for the authors to improve the paper (not meant as true weaknesses):

- title: remove the space between "optimum" and the question mark
- line 54: "the best approximation property (BAP), which guarantees the existence of an optimal solution" -> Is this the definition of BAP or just a consequence? Please make this clear. You might want to define BAP in a proper definition environment.
- lines 57-79: for readers unfamiliar with sparsity in neural networks, it is quite hard to understand your contributions without reading the "notations" part later. You should spend some efforts in explaining terms like "(structured) sparse networks", "fixed sparsity level" vs. "fixed sparsity pattern", etc. already in the introduction. I think "sparsity level" is only implicitly defined very late in the paper.
- line 67: Here (and at some other places in the paper) you use the term "learning problem" as a synonym to "training problem". Some people understand "learning problem" as the problem to minimize the generalization error, as opposed to the "training problem", which only aims to minimize the training error. At least in the finite domain case you are definitely in the latter regime, so I suggest to use "training problem" consistently.
- Tables 1 and 2: the relation to the sparsity constraints is not really clear in the tables. In particular, what does the "sparse" adjective in brackets in the architecture column mean? Why is it in brackets?
- caption of table 2: there is an extra space after the opening bracket
- line 124: it is a bit weird that the "notations" section is part of the "related work" section.
- line 195: contains -> contain
- line 261: valed -> valued
- line 293: a bit redundant "many other domains such as [...] and much more".
- line 301: "(in the whole paper we naturally assume B > 0)" -> state this earlier in the paper when you use B for the first time.
- line 303: provide a reference to the proof of this theorem in the appendix.
- line 312: what you call "plain" here is what I would call "fully connected". You should maybe use this term here and elsewhere in the paper.

**Questions:**

- Is there any chance to obtain a hardness result for the computational problem considered in Section 3.3? NP-hardness? Hardness for the existential theory of the reals (which is always a good candidate if the best known solution uses quantifier elimination)? See related results for NN training: https://arxiv.org/abs/2102.09798; https://arxiv.org/abs/2204.01368.
- Pushing the relation to the existential theory of the reals even further, I find it very interesting that you obtain such a sharp difference between the scalar-valued output case and the multi-dimensional output case (Section 3.4). The same seems to be true for the training complexity of such shallow networks: while scalar-output networks can be trained with a combinatorial search algorithm, certifying that the problem is in NP, it turns out that training a shallow network with multi-dimensional output is ER-complete and therefore much harder (https://arxiv.org/abs/2204.01368). Again I wonder: is there any relation to your results?

**Limitations:**

The paper properly states under which circumstances (mathematical assumptions) the results are valid.

---

> ### Author Rebuttal · Authors · 2023-08-06
>
> We thank the referee for his/her comments and questions.
>
> **Comments on weakness** We really appreciate the comments on the overall organization and presentation of the paper and will implement them correspondingly in the final version.
>
> Regarding the *limitation to 2-layers only*, all our *negative results* (sufficient conditions for non-closedness, based on Theorem 3.1) are valid for any depth.  Several technical lemmas for our *positive results* (e.g.  Lemma C1 and Lemma C2 in the Appendix) also hold for MLPs with arbitrary depths. These results could be re-used to analyze more completely the deeper case, which is however left for future work. Moreover, our positive results apply to non-scalar outputs and take into account the effect of sparsity constraint, while (to the best of our knowledge) existing positive closedness results were limited to one-hidden-layer, scalar-valued outputs, with no support constraint.
>
> **Answer to question**
>
> 1. The question is indeed interesting. We found several links between the papers mentioned by the referee and our setting in Section 3.3. Indeed, in the proof of the main result of the first paper, the constructed ``hard'' neural network does not have all the connections (i.e., the support is not full), it is assumed to have the identity activation function (Theorem 2) and all biases can be assumed to be zero (Observation 5). This is very similar to the setting of sparse matrix factorization. Nevertheless, it remains non-trivial to adapt the techniques used in the articles mentioned by the referee to the problem of deciding on the closedness of the semi-algebraic set $\mathcal{L}_\mathbf{I}$. Still, we believe this interesting question is worth investigating further.
>
> 2. We think the two results are somewhat related. In fact, our proof of Theorem 3.4 uses the normalization technique from Algorithm~1 [1]. The exact same algorithm is used in the paper mentioned by the referee to argue that the training of scalar-valued output, one-hidden layer NNs is NP (if the input and hidden-layer dimensions are constant). We believe it is interesting to exploit this observation and further study the separation between the cases of scalar-valued and vector-valued output.
>
> **References**
>
> [1] R. Arora, A. Basu, P. Mianjy, A. Mukherjee, Understanding Deep Neural Networks with Rectified Linear Units, International Conference on Learning Representations, ICLR, 2018.

---

> > ### Comment · Reviewer_7u6K · 2023-08-13
> >
> > I thank the authors for sharing their thoughts in the rebuttal and answering my questions. I remain curious about the connections to results in training complexity and hope future work will generate more insights on this. I continue to vote for acceptance of this paper.

---

### Official Review · Reviewer_5AbX · 2023-07-07

**Soundness:** 4 excellent
**Presentation:** 4 excellent
**Contribution:** 3 good
**Rating:** 7
**Confidence:** 3

**Summary:**

The paper studies the existence of an optimal solution to the objective of training a ReLU neural network under certain sparsity patterns. They consider two topological properties of the input function space (the best approximation property, and closedness). They provided series of results on scalar-valued neural networks, shallow neural networks, neural network with one hidden layer. Specifically, they provide necessary and sufficient conditions on the sparsity (e.g., fixed sparsity level or pattern) of the neural network to guarantee existence of an optimal solution to the training loss function.

**Strengths:**

The paper studies an interesting problem which relates to the stability of the neural network training due to the non-existence of an optimal solution given a sparsity pattern. The paper covers a good literature review and stand themselves very well compared to prior works. Their contribution is clear.

The title of the paper is very related to the goal of the paper. The paper is coherent in studying the problem of interest.

The paper is written clearly. Each theorem/proposition is followed by lemma/corollary and informal proof sketch which makes it very easy to follow. For examples, the paper contains a numerical example very early on in the paper to further give intuition on the importance of the problem and a plausible scenario where an optimal solution does not exist.

The results provide new insights on if a neural network will have an optimal solution. Table 1 and 2 clearly state how their work differs from prior work.

**Weaknesses:**

It would be nice in the main paper to provide more tangible examples on what sparsity examples on what architectures may/may not result in existence of an optimal solution. NeurIPS has a broad range of audiences from theory and application (addition of this can attract more application-based deep learning researchers).


- minor: corollary 3.1, valed --> valued

**Questions:**

- For corollary 3.1, what is the dimension of the hidden layer?

- Could you clarify in simple words the term number 2 on line 305 (addition of this will improve clarity).

**Limitations:**

Addressed properly.

---

> ### Author Rebuttal · Authors · 2023-08-06
>
> We thank the reviewer for his/her comments on our work.
>
> **Comment on the weakness** Our work allows to answer the following question: if the supports of the weight matrices are randomly sampled from a distribution, what is the probability that the corresponding training problem potentially admits no optimal solutions? While simple, this setting does happen in practice since random supports/binary masks are considered a strong and common baseline for sparse DNNs training [1].
>
> Indeed, thanks to Theorem 3.1, if $\mathcal{L}\_\mathbf{I}$ is not closed then the support is *bad*. Although the algorithm of Lemma 3.3 to *decide* if $\mathcal{L}\_\mathbf{I}$ is closed is not polynomial, for one-hidden-layer NNs, there is a polynomial algorithm to *detect* non-closedness: if the support constraint is *locally similar* to the LU structure (precisely, if it satisfies the condition of Theorem 4.20 of [2]), then $\mathcal{L}\_{\mathbf{I}}$
> is not closed. The resulting detection algorithm can have false positives (i.e., it can fail to detect more complex configurations where $\mathcal{L}\_\mathbf{I}$ is not closed) but not false negative.
>
> When testing it on a one-hidden layer ReLU network with two $100 \times 100$ weight matrices, drawing uniformly at random two supports of respective cardinality
> $|I_1| = 3000$ and $|I_2|=2000$ (i.e. 30\% of nonzero coefficients on the first layer, and 20\% on the second one) and averaging over 100 draws, the algorithm estimates the probability of "bad" supports to 85\%. For sparser random supports (when $|I_1|=|I_2|=2000$), the estimated probability is nearly 100\%.
>
> We will add a brief description of these consequences of our work in the final version, thank you for this opportunity.
>
> **Answer for questions**
>
> 1. The dimension of the hidden layer is arbitrary (there is no assumption on it). This will be clarified in the final version.
>
> 2. An equivalent (and probably simpler) way to state the second point is: for each fixed binary diagonal matrix $D$, the set $\\{W_2DW_1 \mid \text{supp}(W_1) \subseteq I_1, \text{supp}(W_2) \subseteq I_2\\}$ is closed. This will be clarified in the final version.
>
> **References**
>
> [1] S. Liu, T. Chen, X. Chen, L. Shen, D.-C. Mocanu, Z. Wang, M. Pechenizkiy, The Unreasonable Effectiveness of Random Pruning: Return of the Most Naive Baseline for Sparse Training, International Conference on Learning Representations, ICLR, 2022.
>
> [2] Q.-T. Le, E. Riccietti, R. Gribonval. Spurious Valleys, NP-hardness, and Tractability of Sparse Matrix Factorization With Fixed Support. SIAM Journal on Matrix Analysis and Applications.

---

> > ### Comment · Reviewer_5AbX · 2023-08-13
> > **Reviewer after Rebuttal**
> >
> > I thank the authors for additional clarifications on my questions. I have read their response and recommend acceptance (keep my score).

---

### Author Rebuttal · Authors · 2023-08-06

Dear reviewers,

In this global response, we attach a pdf file containing figures for our rebuttal.

---

### Decision · Program_Chairs · 2023-09-21

**Decision:**

Accept (poster)

**Comment:**

This paper shows that optimization problems involving deep networks with certain sparsity patterns do not always have optimal parameters, and that optimization algorithms may then diverge. The results are interesting and all reviewers recommended acceptance. Some of the reviewers have pointed out some typos, and I suggest that the authors should revise the paper accordingly.